# Insights from Earth Map: Unraveling Environmental Dynamics in the Euphrates–Tigris Basin

Ayhan Ateşoğlu [1,*], Mustafa Hakkı Aydoğdu [2], Kasım Yenigün [3], Alfonso Sanchez-Paus Díaz [4], Giulio Marchi [4] and Fidan Şevval Bulut [1]

1   Department of Forestry Engineering, Bartin University, 74100 Bartin, Türkiye; fbulut@bartin.edu.tr
2   Department of Agricultural Economics, Faculty of Agriculture, Harran University, 63100 Şanlıurfa, Türkiye; mhaydogdu@hotmail.com
3   General Directorate of Combating Desertification and Erosion, 06100 Ankara, Türkiye; kyenigun@hotmail.com
4   Office of Climate Change, Biodiversity and Environment, Food and Agricultural Organization of the United Nations, 00154 Rome, Italy; alfonso.sanchezpausdiaz@fao.org (A.S.-P.D.); giulio.marchi@fao.org (G.M.)
*   Correspondence: aatesoglu@bartin.edu.tr

**Abstract**

The Euphrates–Tigris Basin is experiencing significant environmental transformations due to climate change, Land Use and Land Cover Change (LULCC), and anthropogenic pressures. This study employs Earth Map, an open-access remote sensing platform, to comprehensively assess climate trends, vegetation dynamics, water resource variability, and land degradation across the basin. Key findings reveal a geographic shift toward aridity, with declining precipitation in high-altitude headwater regions and rising temperatures exacerbating water scarcity. While cropland expansion and localized improvements in land productivity were observed, large areas—particularly in hyperarid and steppe zones—show early signs of degradation, increasing the risk of dust source expansion. LULCC analysis highlights substantial wetland loss, irreversible urban growth, and agricultural encroachment into fragile ecosystems, with Iraq experiencing the most pronounced transformations. Climate projections under the SSP245 and SSP585 scenarios indicate intensified warming and aridity, threatening hydrological stability. This study underscores the urgent need for integrated water management, Land Degradation Neutrality (LDN), and climate-resilient policies to safeguard the basin's ecological and socioeconomic resilience. Earth Map is a vital tool for monitoring environmental changes, offering rapid insights for policymakers and stakeholders in this data-scarce region. Future research should include higher-resolution datasets and localized socioeconomic data to improve adaptive strategies.

**Keywords:** Euphrates–Tigris Basin; climate change; LULCC; land degradation; Earth Map

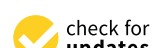

## 1. Introduction

Climate change and environmental degradation are among the most pressing challenges of the 21st century, significantly affecting water resources, land use, and vegetation dynamics [1]. Rising global temperatures, changing precipitation patterns, and the increasing frequency of extreme weather events are reshaping ecosystems and disrupting human livelihoods [2,3]. These changes are particularly evident in river basins, which serve as the lifeline of many regions and provide essential resources for agriculture, industry, and settlements [4,5]. Analyzing environmental changes at the basin level is particularly important for effective management and sustainable development [6]. River basins function

as interconnected systems in which climate, land use, water availability, and vegetation interact in complex ways [7]. Changes in many ecological criteria at the basin scale can have cascading effects throughout the ecosystem. Therefore, monitoring these dynamics at a comprehensive basin-wide scale will enable better resource allocation, informed policy decisions, and targeted conservation efforts. Furthermore, large basins are located across the borders of multiple countries and should be assessed holistically. This makes them critical to regional cooperation and stability [8]. Many of the world's major river basins, such as the Tigris–Euphrates, Nile, and Mekong, are shared by many nations, requiring regional collaborations [9–11]. This basin-level monitoring and management approach supports sustainable land use and land cover development, particularly in regard to climate change [12].

The Euphrates–Tigris Basin is a critical transboundary water system supporting diverse ecological functions in the Middle East. The basin covers parts of Türkiye, Syria, Iraq, and Iran and contains vital wetlands, river ecosystems, and a problematic ecosystem dependent on seasonal water flows [13]. From a hydrological perspective, the Euphrates and Tigris rivers provide essential freshwater resources to a region characterized by arid and semi-arid climates. The rivers originate in the highlands of Türkiye and flow downstream through Syria and Iraq, where they sustain agriculture, urban populations, and industrial activities [14]. The sensitive ecosystem of the basin has recently been exposed to the adverse effects of climate change [15]. The Euphrates–Tigris Basin is of high socioeconomic importance as it forms the backbone of agriculture, energy production, and livelihoods for millions of people [9]. Therefore, integrated basin management, the development of adaptation strategies, and monitoring are essential to reduce the effects of climate change.

The Euphrates–Tigris Basin is increasingly affected by climate change due to increasing temperatures, changing precipitation patterns, higher evapotranspiration rates, higher occurrence of extreme events, etc. [16]. In parallel, regional instability, changes in land use, and land degradation in the basin due to the expansion of agriculture and urbanization further exacerbate environmental challenges. These land degradation and use changes disrupt natural hydrological cycles, placing additional pressure on a region already under water stress. Vegetation degradation in the basin is another major concern as it directly affects ecosystem services [17]. Loss of natural vegetation also contributes to desertification and increased sand and dust storms [15]. Given the increasing challenges posed by climate change, land use changes, and vegetation degradation, a comprehensive monitoring system is essential for the sustainable management of the Euphrates–Tigris Basin. A data-driven approach will enable decision-makers to assess long-term trends, detect early warning signs, and implement adaptive strategies to reduce environmental risks.

Given these challenges, integrating climate, land use, vegetation, and water data in the basin provides a holistic understanding of the region's ecological transformations. In this context, future climate projections based on Shared Socioeconomic Pathways, particularly SSP245 (a stabilization scenario) and SSP585 (a high-emissions scenario), offer critical insights into anticipated changes in temperature and precipitation across the Euphrates–Tigris Basin. These projections suggest increased water scarcity, heightened stress on vegetation, and a greater likelihood of extreme climatic events, including droughts, sandstorms, and dust storms [18]. Integrating scenario-based climate modeling into environmental monitoring frameworks strengthens the capacity to assess long-term risks and to develop adaptive management strategies suited to a range of possible future climate conditions [19]. In recent years, remote sensing (RS) and geographic information systems (GIS) have been the most advanced tools used for this purpose. RS&GIS integration facilitates informed policy decisions by providing more accurate climate impact assessments, land degradation mapping, vegetation monitoring, and water resources management [20,21]. In this context,

open-access global datasets facilitate data-driven decision-making for sustainable resource management by providing valuable insights into spatial and temporal changes [22,23]. Datasets from sources such as the National Aeronautics and Space Administration (NASA), European Space Agency (ESA), and global hydrological models provide valuable information on climate trends, precipitation patterns, land cover changes, vegetation indices, and water availability. These datasets facilitate large-scale analysis by enabling comparisons across regions and time periods, such as major river basins [24,25]. This technological level and big data management make it easier and more explainable for stakeholders in large basins, allowing them to identify climate-related changes, evaluate the impact of land use changes, and assess ecological stress levels using satellite images and geographic analysis. Additionally, open-access data ensure transparency and sustainability, support collaborative decision-making, and increase the capacity to develop evidence-based strategies for climate adaptation and resource management in the basin [26]. One of these platforms, Earth Map (earthmap.org), is a powerful open-access tool for monitoring environmental changes. Developed by the Food and Agriculture Organization (FAO) in collaboration with Google Earth Outreach, Earth Map allows users to analyze climate, land use, vegetation, water dynamics, and many other parameters using Google Earth Engine (GEE)'s extensive satellite data archive. The platform simplifies complex analyses, giving researchers and decision-makers access to historical and near-real-time data through a point-and-click interface and without requiring any programming skills [27,28]. Such an approach allows for a large democratization of the application of an open-access environment and climate data to anyone who lacks the time or the know-how to retrieve them and transform them into actionable information.

The purpose of this study is to assess the impacts of climate change on environmental dynamics within the Euphrates–Tigris Basin by integrating multiple geospatial indicators using Earth Map, a global open-access monitoring platform. Through a structured analysis of climate trends, Land Use and Land Cover Change (LULCC), vegetation dynamics, and water balance, this study aims to identify spatial vulnerabilities and long-term ecological transformations in this critical transboundary region. The research aims to access numerical data by utilizing Earth Map's standardized datasets and provides a comprehensive, scalable, and policy-oriented assessment of regional environmental stress. This approach supports evidence-based decision-making for sustainable land and water resource management. It lays a foundation for future research focused on modeling causal relationships and developing adaptive climate strategies.

## 2. Materials and Methods

### 2.1. Study Area

The Euphrates–Tigris Basin is one of the most essential transboundary river systems in the Middle East, spanning parts of Türkiye, Syria, Iraq, and Iran. Originating in the highlands of eastern Türkiye, the Euphrates and Tigris rivers flow through arid and semi-arid lands before merging in southern Iraq and emptying into the Persian Gulf [29]. Covering an area of approximately 930,000 km$^2$, the basin supports agriculture, drinking water, hydroelectric production, and biodiversity, making it a critical natural resource for the region (Figure 1). The climate of the basin varies from humid conditions in the mountainous regions in the upper basin to arid and semi-arid conditions in the lower basin [30]. Seasonal rainfall and snowmelt in the Anatolian Plateaus are important in maintaining river flow. Over the last few decades, the basin has experienced ecosystem stress due to climate change and unsustainable land use practices [31].

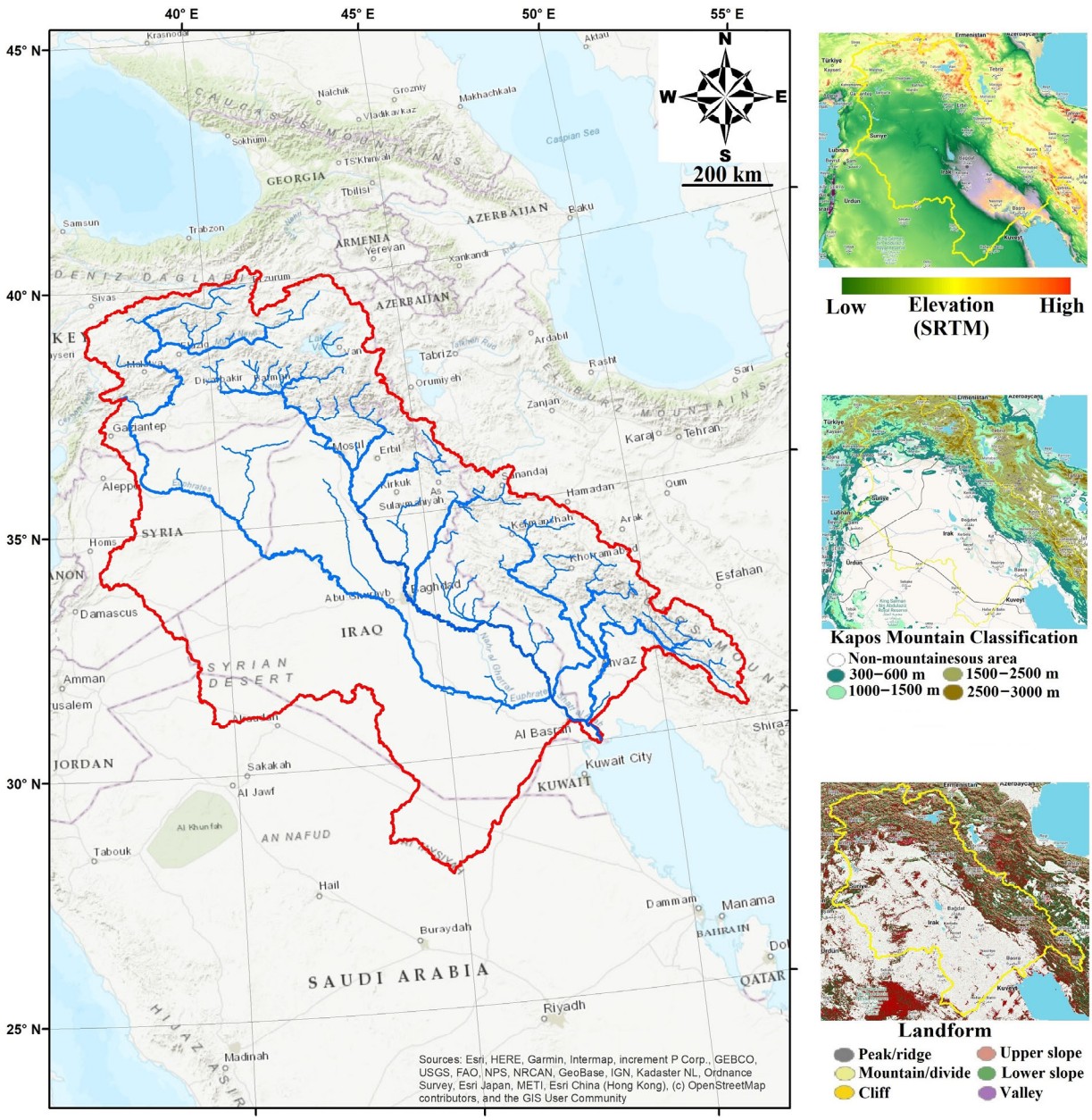

**Figure 1.** Study area (left map from [origin of left map]; three right maps from Earth Map).

Given that the Euphrates–Tigris Basin spans multiple countries with differing socioeconomic conditions and competing water needs, the equitable and sustainable management of transboundary water resources presents a critical challenge [32]. Climate change is expected to further alter the availability and distribution of water in the region, making coordinated management and future resource allocation even more essential [33]. This underscores the need for each riparian state to assume shared responsibility in addressing water-related vulnerabilities. It is imperative to foster cooperative mechanisms that transcend historical disputes, internal political challenges, and conflict dynamics in order to promote resilience and long-term regional stability [34,35]. Accordingly, a comprehensive understanding of the basin's geopolitical context and future scenarios concerning water and food security is vital for formulating effective climate adaptation strategies and advancing sustainable development across the basin.

## 2.2. Earth Map

Earth Observation (EO) data have played a critical role in understanding urgent environmental problems and the impacts of climate change, enabling us to make significant advances in land monitoring and climate assessment. The recent years have been fundamental regarding the amount of EO and climate data freely available to the public and researchers [36]. Technological advances (such as cloud-based storage and processing power) and open-data policies adopted by governments and space agencies have driven improvements in accessibility [37]. Cloud computing technologies and free satellite data are revolutionizing how countries, organizations, academia, and even individuals approach the management of natural resources, including monitoring climate parameters, environmental problems, deforestation, and desertification [38–42]. Since 2016, the Food and Agriculture Organization (FAO) has been developing Earth Map (earthmap.org), a simple and user-friendly interface that provides anyone easy access to many features and datasets in Google Earth Engine (GEE) and other sources that are mostly used for environment and climate analysis and to the GEE powerful cloud computational capacity; the point-and-click platform does not require users to master any coding techniques and is accessible to anyone through a browser and internet access [37].

Earth Map is a web-based application that consists of a map where geographic layers can be easily viewed, and statistics can be generated instantly through its graphical user interface. Earth Map's data are currently divided into 18 thematic groups (layers) covering agriculture, biodiversity, climate, energy projections, greenhouse gas emissions, energy, fire, forestry, geophysics, hydrology, imagery, land use/land cover, Land Degradation Neutrality, protected areas, social, soil, vegetation, and water. The tool allows the user to visualize the layers (maps) with their corresponding descriptions on top of the background maps of Google Maps. More importantly, the user can perform deeper analysis by generating zonal statistics in AOIs that complement the visual information of the maps; spatial data or charts and tables can be easily exported in commonly used formats for further processing. An AI Explain feature has been recently added to describe the statistics in natural language. AOIs can be the boundaries provided in Earth Map, basin borders from the WWF HydroSHEDS dataset, or the user can either draw or import his/her own AOIs as KML, GeoJSON, or shapefile. GEE provides Earth Map with the capacity to run on-the-fly analysis on most of the metrics of the images, such as temperature, precipitation, burned areas, tree cover, drought index, and many others [43,44]. These zonal statistics can be run in seconds on any device, regardless of the computing power of the device, since their processing occurs on the cloud. Statistics can be collected at different temporal aggregations (annual, monthly averages, and monthly time series) and over various time periods. Since statistical analysis is performed on the fly, information can be obtained at a global, regional, or project level (Figure 2). This makes Earth Map a multi-temporal, multi-parametric, and multi-scale geospatial platform for a large number of environmental and climatic data. Earth Map is a web-based and freely accessible application with a graphical user interface (GUI) built on top of the GEE Application Programming Interface (API) to interact with the GEE servers to display the maps and generate statistical results. The Earth Map GUI is responsible for displaying all geographic data (layers and statistics) to the user and presenting them in a comprehensible and usable format. It uses the Google Maps API as its primary interface component. The code structure in Earth Map uses a combination of front-end JavaScript components and server-less NodeJs code to interact with GEE (Figure 3). All details about Earth Map and its comprehensive methodology are presented in [25].

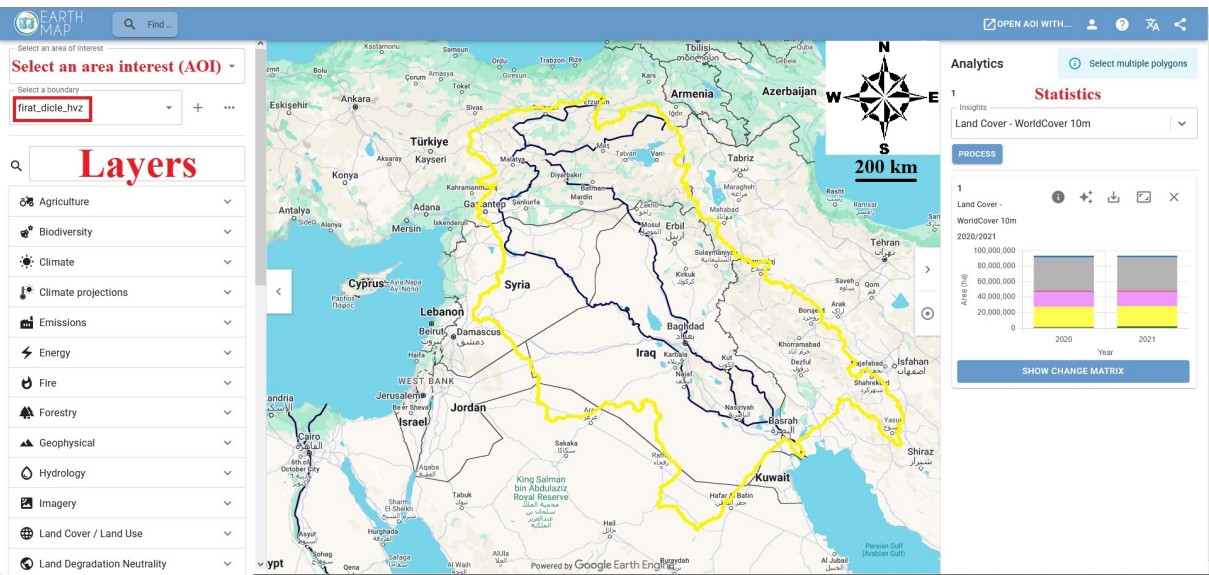

**Figure 2.** Earth Map's interface.

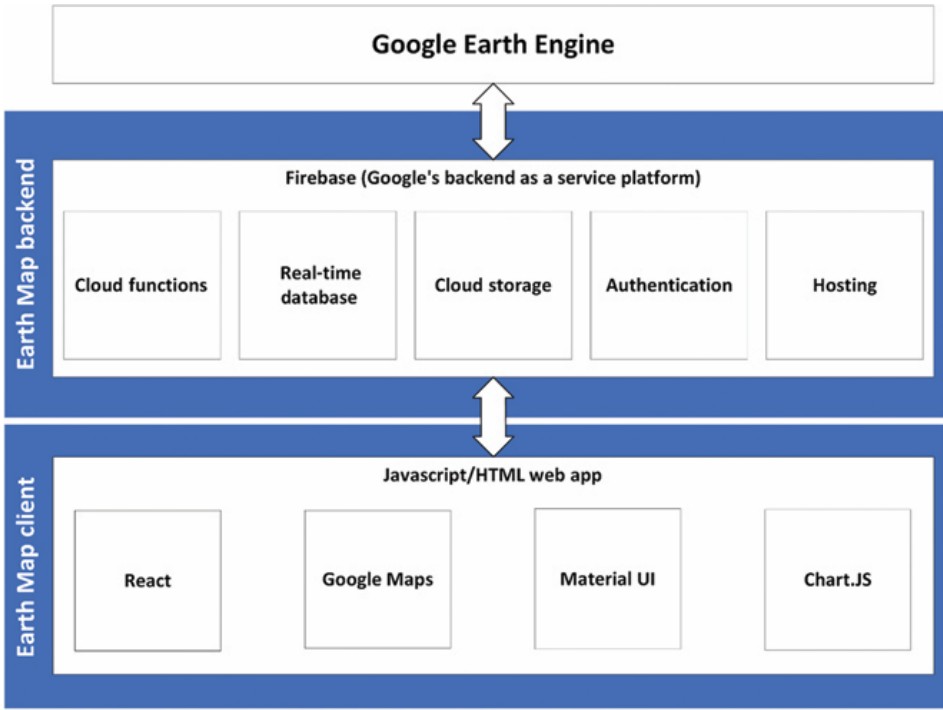

**Figure 3.** Earth Map's software architecture [27].

Through just a click, the user can visualize the same area under analysis in two external platforms:

- a streamlined GEE application called Imagery Comparison (https://earthmap.org/compare.html, accessed on 14 March 2025) with access to more geospatial products and side-by-side image comparison.
- the FAO AgroInformatics Geospatial Platform (https://data.apps.fao.org, accessed on 5 April 2025) with additional resources and analytical capabilities.

Earth Map received a Google Geo for the Good Impact Award in 2024 [45].

Every dataset in Earth Map comes with an Information button through which the user can first access a short description of the data, from which, according to the data involved, they can directly access the source data, the processed data, and the GEE asset and script.

Earth Map includes a full Help Center with how-to guides, examples, tutorials, FAQs, publications, and a Contact Us form to report suggestions, bugs, or data requests.

In this study, eight distinct datasets available through the Earth Map platform were utilized to comprehensively analyze the environmental dynamics of the Euphrates–Tigris River Basin.

### 2.3. Datasets

2.3.1. Global Ecological Zones

Ecological zones are typically classified based on long-term climate patterns, vegetation, and physiographic characteristics. The FAO Global Ecological Zones (GEZ) dataset provides a standardized global framework for ecological zoning, primarily developed to support forest resource assessments and climate-related ecosystem monitoring [46,47]. The classification is based on a combination of climatic parameters—such as temperature and precipitation regimes—alongside natural vegetation types, elevation, and biogeographic distribution. The GEZ mapping methodology incorporates climate normals (typically 30-year averages), derived from global climatological datasets such as CRU and WorldClim, and uses a combination of expert knowledge and cluster analysis techniques to delineate zones. The resulting ecological zones represent relatively homogeneous areas in terms of vegetation structure, climate conditions, and ecosystem functioning. These zones are periodically updated to reflect changes due to climate trends, improved input data, or methodological refinements. In the latest version, 20 primary zones are defined globally, including boreal, temperate, subtropical, and tropical domains, with further subdivision based on humidity levels (e.g., dry, moist, wet, and mountain). The statistical validation of GEZ classifications is performed using remote sensing data, ground observations (where available), and spatial overlays with independent ecological classifications. While not based on a statistical test, the methodology includes iterative calibration and peer review to ensure ecological and climatological consistency across global regions. In the context of climate change analysis, the GEZ dataset provides a valuable spatial reference for evaluating ecosystem shifts, identifying zones at risk of transition, and assessing the vulnerability of specific ecological domains to climatic variability and extremes. Comparing recent observational data against historical GEZ boundaries helps highlight areas experiencing significant bioclimatic shifts, offering an evidence base for adaptive land use planning and ecosystem resilience strategies [48]. In the context of the Euphrates–Tigris Basin, GEZ data—by delineating ecologically homogeneous zones—provide a valuable framework for assessing ecosystem vulnerability, interpreting and monitoring bioclimatic changes and climate sensitivity, and informing adaptive land management strategies.

2.3.2. Aridity Index (AI)

To calculate the AI on an annual basis, the 8-day potential evapotranspiration (PET) data from the MOD16A2 MODIS/Terra Net Evapotranspiration 8-Day L4 Global 500 m Version 6 product are first aggregated. For each year in the study period (2001–2020), all 8-day PET values are summed to produce the annual total PET per pixel. This process obtains an annual PET dataset with a spatial resolution of 500 m. Precipitation data are obtained from the ERA5-Land dataset provided by the European Centre for Medium-Range Weather Forecasts (ECMWF). Hourly precipitation values are aggregated into annual totals for each corresponding year. Since the native spatial resolution of ERA5-Land is approximately 9 km, the annual precipitation data are resampled to a 500 m resolution using bilinear interpolation so that the spatial resolution matches that of the MOD16A2 PET dataset. This allows for the pixel-wise calculation of the Aridity Index, defined as the ratio of annual precipitation to annual potential evapotranspiration

(AI = P/PET), for the years 2001 to 2020. The median AI values for the two decadal periods (2001–2010 and 2011–2020) are calculated to examine temporal shifts in aridity. A difference map is then generated to visualize spatial changes in aridity conditions over time [49]. This study employed the Aridity Index (AI) to evaluate spatial and temporal variations in drought conditions across the Euphrates–Tigris Basin over the 2001–2020 period. Calculated as the ratio of annual precipitation to potential evapotranspiration, the AI enabled quantifying drought intensity and identifying transitions between aridity classes. As a diagnostic tool, the AI provided critical insights into the influence of both climatic variability and anthropogenic land management practices on drought dynamics. Moreover, it facilitated the detection of emerging drought risks, especially for regions that play a vital role in sustaining the basin's hydrology.

### 2.3.3. Precipitation/Temperature

The two most important indicators of drought are temperature and precipitation [50]. However, looking at anomalies and changes, not just raw values, provides a long-term perspective on climate change and helps inform adaptation strategies. Anomalies help detect and respond to ongoing droughts, while average changes help plan for the future by showing how the climate is evolving [51]. The "Total Precipitation Change", "Mean Temperature Change", "Total Precipitation Anomalies", and "Mean Temperature Anomalies" products are derived from processing the European Centre for Medium-Range Weather Forecasts (ECMWF) ERA5 atmospheric reanalysis of the global climate product. The Total Precipitation Change map represents the change in the average total precipitation per year for the whole period, while the Mean Temperature Change represents the change in the average mean temperature per year for the entire period (by year). The Precipitation Anomalies and the Mean Temperature Anomalies are calculated by building an image with the average of the Annual Precipitation and Mean Temperature values for the region of interest for the whole period (1979–2019) and comparing it to the average Annual Precipitation value and the Mean Temperature value (per pixel) of the years 2014–2019. The anomalies map shows the percentage deviation (per pixel) between 2014 and 2019. The pixel size is 0.25 deg (approx. 28 km at the equator), and the period of observations is 1979 to the present (5 days of lag time for processing). Monthly aggregates are calculated based on the ERA5 hourly values of each parameter [52]. In addition, the NEX-GDDP-CMIP6 dataset provides high-resolution and bias-corrected global climate projections downscaled from CMIP6 General Circulation Models (GCMs), developed for the IPCC Sixth Assessment Report. It includes daily data based on ScenarioMIP, which runs under two Tier 1 Shared Socioeconomic Pathways (SSPs). The dataset is created by applying statistical downscaling and bias correction to global model outputs to better capture local climate variations and topographic influences [53,54]. In this study, temperature and precipitation data were used to assess both historical and projected climate trends and their impact on drought risk and water resource sustainability in the Euphrates–Tigris Basin. These datasets can identify increasing climate instability, where wet and dry years alternate, threatening the reliability of the basin's ecosystem cycle. Furthermore, future projections based on the NEX-GDDP-CMIP6 scenarios highlight significant warming and potential increases in drought under both moderate- and high-emission pathways. Together, these climate indicators are essential for understanding past and future drought dynamics, supporting the development of adaptation strategies, and informing sustainable ecosystem management under changing climate conditions.

### 2.3.4. Land Cover and Land Use Change

In this study, the Global Land Cover and Land Use Change 2000/2020 dataset, developed by the University of Maryland's Global Land Analysis and Discovery (UMD GLAD) laboratory, is used as the primary source for land use and land cover (LULC) information. This dataset is particularly suitable for assessing the long-term impacts of climate change, as it provides spatially consistent high-resolution data (30 m resolution) on global land cover transitions over the two-decade period from 2000 to 2020 [55]. A key innovation of the UMD GLAD dataset is its direct change detection methodology, which differs from traditional "post-classification comparison" approaches. Instead of independently classifying land cover for each year and comparing the results, the GLAD dataset applies a bi-temporal supervised classification using paired Landsat imagery from 2000 and 2020. This method employs a consistent set of training data and classification algorithms to directly detect and label land cover transitions at the pixel level, which minimizes the compounding of classification errors across time steps [56]. The classification process involves the use of random forest machine learning algorithms, trained on thousands of globally distributed reference samples, to distinguish among 10 major land cover classes (e.g., forest, cropland, grassland, urban, and shrubland) [57]. Spectral indices, such as the Normalized Difference Vegetation Index (NDVI), and moisture and texture metrics are extracted from Landsat surface reflectance imagery and used as inputs to the classifier [58]. Importantly, the dataset also incorporates ancillary data layers—such as elevation, slope, and climate variables—to enhance classification accuracy in ecologically complex areas. In addition to producing a static land cover map for each year, the dataset provides a transition matrix that specifies which land cover classes changed and the direction and magnitude of these changes. For example, it captures transitions such as "forest to agriculture", "grassland to urban", or "wetland to barren", which are particularly relevant in the context of climate change and land degradation studies [55]. These matrices enable detailed spatial and statistical analysis of LULC dynamics at both regional and global scales. By integrating high-resolution remote sensing data with advanced machine learning and change detection algorithms, the GLAD dataset allows for more reliable detection of long-term land cover changes, supporting robust assessments of human and climate-induced landscape transformations. This study utilized the GLAD dataset to analyze LCLUC, focusing on key land cover classes, such as wetlands, cropland, and built-up areas, in order to examine how related socio-environmental processes have influenced land use dynamics in the Euphrates–Tigris Basin. The LCLUC analysis highlighted the complex interplay between environmental stressors—particularly drought—and human adaptive responses, revealing significant agricultural and urban expansion into ecologically sensitive areas, such as semi-arid zones and wetlands. These findings offer important insights into the implications of land use transformations for long-term environmental sustainability and resource management under conditions of climatic and political stress.

### 2.3.5. Normalized Difference Vegetation Index, Potential Evapotranspiration, and Water Deficit

Vegetation monitoring is a critical parameter in assessing the impacts of drought, land management practices, and climate variability within a river basin [59]. Among various remote sensing tools, the Normalized Difference Vegetation Index (NDVI) is one of the most widely used indices for tracking both spatial and temporal variations in vegetation cover [60]. The NDVI is calculated using the formula: NDVI = (NIR − RED)/(NIR + RED), where NIR is the near-infrared reflectance and RED is the red reflectance, both derived from multispectral satellite imagery. NDVI values range from −1 to +1, with higher values indicating denser and healthier vegetation. Positive trends in the NDVI typically reflect

improved vegetation health or increased greening, while negative trends suggest vegetation degradation or stress [61]. In this study, NDVI data were obtained from the MOD13Q1.061 Terra Vegetation Indices 16-Day Global 250 m product, covering the period from 2000 to the present.

Another key variable in environmental monitoring is potential evapotranspiration (PET), which estimates the amount of water that would evaporate and transpire from a surface with sufficient water supply. PET is influenced by temperature, solar radiation, wind speed, and humidity. Although various models exist to estimate PET (e.g., Penman–Monteith and Hargreaves), in this study, PET values were sourced from the MOD16A2 MODIS/Terra Net Evapotranspiration 8-Day L4 Global 500 m (Version 6) dataset, which uses the Penman–Monteith equation, modified for satellite inputs:

$$PET = \frac{\Delta(R_n - G) + P_a c_p \frac{e_s - e_a}{r_a}}{\Delta + \gamma\left(1 + \frac{r_s}{r_a}\right)} \tag{1}$$

where $R_n$ is net radiation, GG is soil heat flux, $\rho_a$ is air density, $c_p$ is specific heat of air, $e_s - e_a$ is vapor pressure deficit, and $r_s$ and $r_a$ are surface and aerodynamic resistances, respectively. PET serves as a reliable indicator of atmospheric demand for moisture and is particularly useful for identifying ecosystem stress in arid and semi-arid environments [62]. To assess water availability and moisture stress more directly, the Water Deficit (WD) is calculated by subtracting Actual Evapotranspiration (ET) from PET: Water Deficit = PET − ET. A positive Water Deficit indicates that atmospheric water demand exceeds available soil moisture, which can lead to drought, vegetation stress, and land degradation [63]. ET data are also derived from the MOD16A2 product (2000–2024), which incorporates meteorological and remote sensing variables to estimate actual plant water use. For temporal analysis, all time series data (NDVI, PET, and WD) were processed using statistical methods such as linear regression and Mann–Kendall trend tests to detect the significance and direction of long-term trends ($p$-values < 0.05 considered statistically significant). Seasonal trend decomposition and z-score normalization were also applied, where necessary, to understand interannual variability and anomalies. Collecting these indicators—NDVI for vegetation condition, PET for climatic water demand, and WD for water stress—provides a comprehensive understanding of the ecological responses to climate change within the river basin. This integrated approach supports the development of sustainable water resource management strategies and adaptation measures for ecosystem and community resilience [64,65].

Together, these variables enabled a comprehensive evaluation of vegetation health, climatic stressors, and changing water availability under ongoing climate change, supporting the identification of vulnerable areas and informing sustainable resource management strategies within the basin. Using average values provides a baseline understanding of typical conditions, while change maps reveal spatial shifts in vegetation productivity, evapotranspiration, and water stress over time. Anomaly analyses are crucial for detecting short-term deviations from normal conditions, helping to identify extreme events such as droughts or unusually wet periods. Trend analyses, including linear regression and statistical significance testing, allow for detecting long-term directional changes, offering critical insights into how ecosystems are evolving in response to climate variability and anthropogenic pressures. Collectively, these analytical approaches enhance the temporal and spatial resolution of environmental monitoring and support evidence-based decision-making for adaptation and resilience planning.

### 2.3.6. Land Productivity Dynamics

To better understand land degradation, climate change impacts, and land management practices' effectiveness, it is essential to monitor land productivity over an extended period, primarily through satellite-based observations of vegetation dynamics. Land Productivity Dynamics (LPD) is a key indicator for evaluating the health and productive capacity of land ecosystems, as it captures long-term changes in vegetation growth patterns and ecosystem function [66]. It is also one of the three sub-indicators used to assess progress toward achieving Land Degradation Neutrality (LDN), alongside land cover change and soil organic carbon trends [67].

LDN aims to balance land degradation with restoration or sustainable land management efforts, ensuring that the net quantity and quality of land resources remain stable or improve over time. According to the United Nations Convention to Combat Desertification (UNCCD), LDN is achieved when:

LDN = ($LD_{baseline}$ − $LD_{current}$) + Gains ≥ 0, where $LD_{baseline}$ is the area of degraded land at a reference year (usually 2000), $LD_{current}$ is the area currently degraded, and Gains refer to improvements in productivity or restored areas.

LPD is primarily assessed using long-term NDVI time series, typically derived from moderate-resolution satellite products, such as MODIS or AVHRR. To analyze LPD trends over time, statistical techniques such as non-parametric trend analysis (e.g., Mann–Kendall test) and Theil–Sen slope estimation are applied to each pixel in the NDVI time series. This enables the detection of monotonic trends in vegetation productivity without assumptions of normal distribution or linearity. The direction and magnitude of these trends are then classified into five productivity trajectories: increasing, stable, early signs of decline, declining, and degraded. This method—originally developed by Ivits and Cherlet [68]—was later adopted in global assessments, such as the World Atlas of Desertification [69]. The Earth Map platform evaluates land productivity using 16-year rolling periods (e.g., 2001–2016, 2002–2017, . . ., up to 2009–2024). Each time window is assessed for significant positive or negative trends in the NDVI, benchmarked against the initial condition in the baseline year. Pixels that show a consistent negative slope in NDVI values are flagged as areas of declining productivity, potentially indicating land degradation. This approach allows for robust land condition monitoring, especially when ground data are limited or unavailable. By integrating LPD trends into LDN monitoring frameworks, policymakers and land managers can identify priority areas for intervention, measure the effectiveness of restoration efforts, and align national strategies with global goals such as Sustainable Development Goal 15.3—"By 2030, combat desertification, restore degraded land and soil, and strive to achieve a land degradation-neutral world" [70].

Land Productivity Dynamics (LPD) serves as a vital indicator for detecting degraded areas, early signs of vegetation decline, and regions exhibiting increased productivity, thereby elucidating the spatial and temporal patterns of land condition changes. Integrating LPD into this analysis enhances the identification of vulnerable ecosystems and facilitates the evaluation of land management effectiveness. Moreover, its application aligns with international frameworks, such as Land Degradation Neutrality (LDN) and Sustainable Development Goal 15.3, which emphasize combating land degradation and promoting sustainable land use. In the context of the Euphrates–Tigris Basin, LPD was employed to systematically monitor and assess long-term trends in vegetation growth and ecosystem functionality, providing a scientifically robust basis for evidence-driven decision-making aimed at land restoration and resilience under ongoing environmental and climatic challenges.

The selected datasets were chosen considering their relationships with key indicators of climate change, land use, vegetation, and hydrological conditions, providing a solid basis for assessing both temporal and spatial patterns in the region (Table 1).

**Table 1.** Overview of Earth Map datasets used in this study.

| Data Name | Short Description | Main Source | Purpose/Target Logic |
|---|---|---|---|
| Global Ecological Zones | The Global Ecological Zones (GEZ) dataset, developed by the FAO, classifies global forests into major ecological types (e.g., tropical rainforest and boreal forest). | Food and Agriculture Organization of the United Nations (FAO) | Climate trends and variability |
| Aridity Index | The United Nations Environment Programme (UNEP) defines drylands according to an Aridity Index (AI), which is the ratio of the average annual precipitation to the potential evapotranspiration. | European Commission, Joint Research Centre | |
| Precipitation/Temperature | The products are derived from processing the European Centre for Medium-Range Weather Forecasts (ECMWF) ERA5 atmospheric reanalysis of the global climate product. | Copernicus Climate Change Service | |
| Land Cover and Land Use Change | The GLAD Global Land Cover and Land Use Change dataset quantifies land use/cover changes from 2000 to 2020 at a 30 m spatial resolution. | GLAD Global Land Cover and Land Use Change | Land Cover and Land Use Change (LCLUC) and gain/loss conversions |
| Potential Evapotranspiration | The potential evapotranspiration (PET) products are derived from the available MODIS Global Terrestrial Evapotranspiration 8-Day Global 1 km time series. | MOD16A2 v006—MODIS/Terra Net Evapotranspiration 8-Day L4 Global 500 m SIN Grid | Vegetation dynamics, climatic water demand, and Land Degradation Neutrality |
| Water Deficit | The Water Deficit product is derived from processing MODIS/Terra Net Evapotranspiration 8-Day L4 Global 500 m version to generate a time series of the Water Deficit. | | |
| Normalized Difference Vegetation Index | The NDVI products are derived from the available Vegetation Indices 16-Day Global 250 m time series. | MOD13Q1 v006 MODIS/Terra Vegetation Indices 16-Day L3 Global 250 m SIN Grid | |
| Land Productivity Dynamics | The dynamics in the land productivity indicator are related to changes in the health and productive capacity. | | |

## 3. Results and Discussions

In this study, the analysis of the Euphrates–Tigris Basin was carried out using Earth Map, focusing on climate trends and variability, LULCC, vegetation, and water monitoring. The findings were evaluated regarding adaptation to climate change and its effects, ensuring sustainable water resources management and integrated basin management strategies. It was emphasized that Earth Map facilitates a comprehensive assessment by using open-access satellite data to monitor environmental trends over time. It is an effective tool that provides valuable insights for decision-makers and stakeholders.

### 3.1. Climate Trends and Variability

According to the GEZ, 26.20% of the basin is located in the Subtropical Mountain System (Figure 4). The Subtropical Mountain System ecozone in the basin plays an essential role in water supply by acting as a natural reservoir regulating hydrological cycles and supporting downstream ecosystems and human communities [71]. A significant part of this ecozone is located in the Anatolian plateaus of Türkiye and is the main source of the Euphrates and Tigris rivers. Other sources are supplied from the Zagros Mountains, located in the same Subtropical Mountain System [30]. Additionally, 39.23% of the basin is in the Subtropical Desert, and 33.42% is in the Subtropical Steppe ecozone. These dryland

ecosystems require a delicate balance to ensure sustainable land and water management [72] and the resilience of local ecosystems and communities [73].

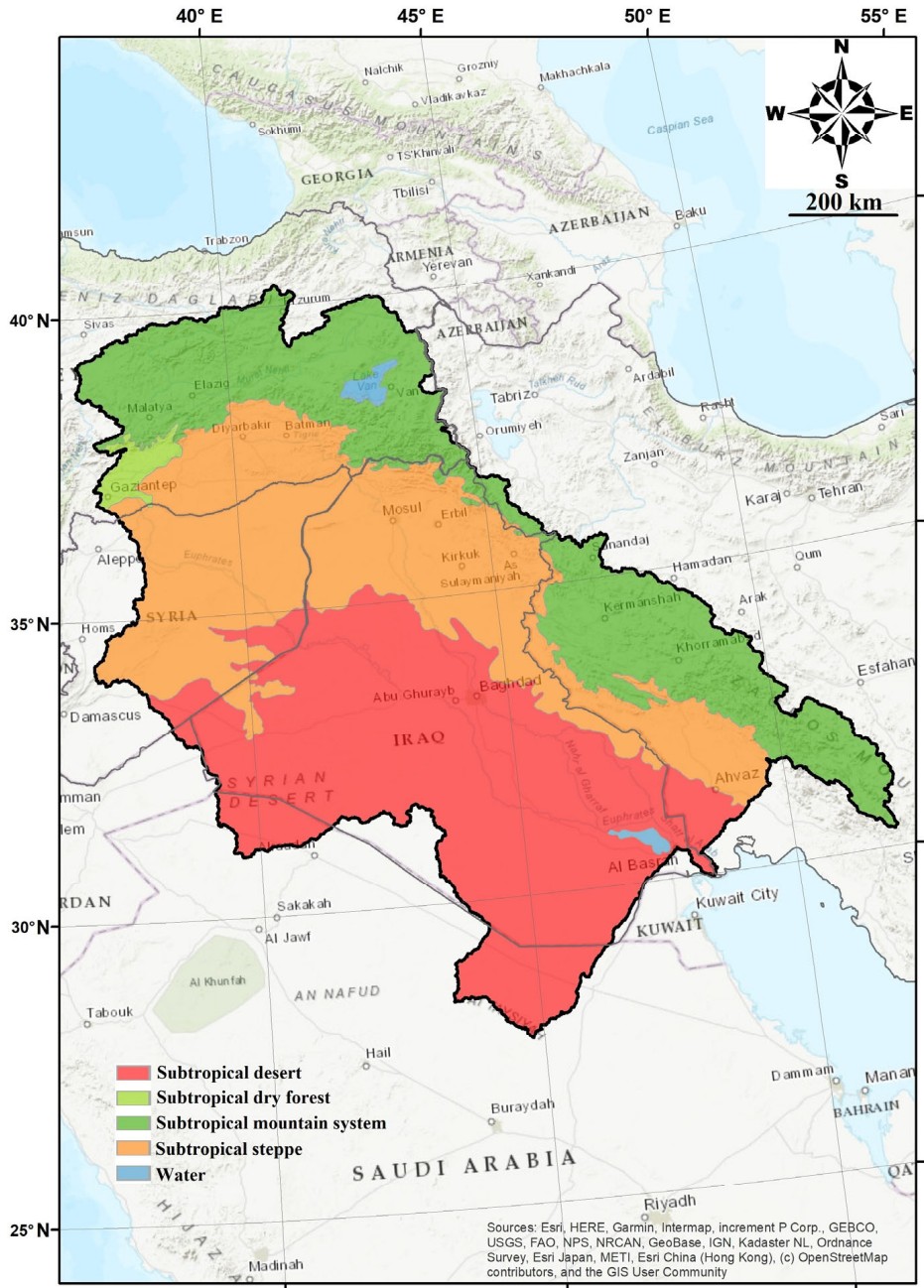

**Figure 4.** Global Ecological Zones of the Euphrates–Tigris Basin.

When the change map was analyzed from the median values of the 2001–2010 and 2011–2020 periods (Figure 5a,b), it was found that the hyperarid class decreased by 4.81%, the arid class decreased by 3.01%, and the non-drylands class decreased by 7.98%. On the other hand, the semi-arid class increased by 9.36% and the dry subhumid class increased by 13.50% (Figure 5c). When aridity changes for the 2001–2020 period are examined, the decrease in arid areas is 7.08%, and the increase in arid areas is 1.64% (Figure 5d).

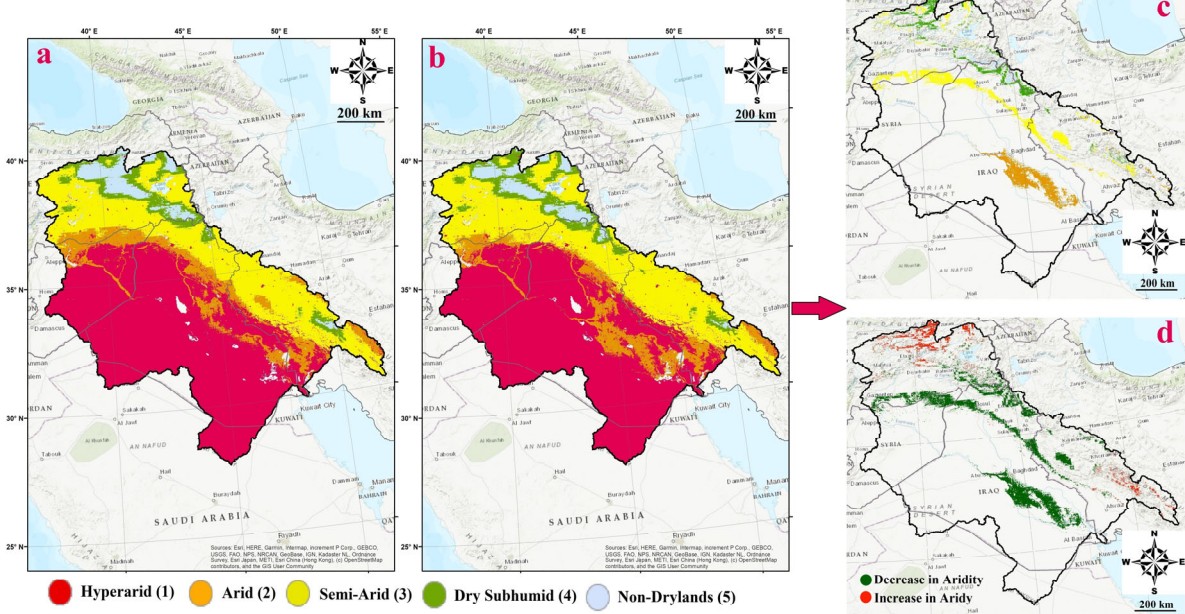

**Figure 5.** Change in Aridity Index (Prec ECMWF Land/PET MODIS); AI of the 2001–2010 period (**a**), AI of the 2011–2020 period (**b**), Comparison of AI classes of changes for 2001–2020 (**c**), and aridity changes for the 2001–2020 period (**d**). The United Nations Environment Programme (UNEP) defines drylands according to an Aridity Index (AI), which is the ratio of the average annual precipitation to the potential evapotranspiration; drylands are lands with an AI of less than 0.65. Drylands are further divided, based on the AI, into hyperarid lands (AI < 0.05), arid lands (0.05 ≤ AI < 0.2), semi-arid lands (0.2 ≤ AI < 0.5), and dry subhumid lands (0.5 ≤ AI < 0.65).

In the basin, it is observed that there is a conversion from the two driest classes (hyperarid and arid) to other classes (semi-arid and dry subhumid) within the AI classes. When the hyperarid and arid conversion regions are examined, geomorphological effects are observed, but specific areas stand out. In particular, drought reduction has been detected in the following regions: the Şanlıurfa (Suruç)-Mardin (Kızıltepe) line within the borders of Türkiye, Münbiç and its surroundings within the Syrian border, Tell Beydar-Tel Hamees and its surroundings, the Tal Afar-Mosul-Kirkuk line in Iraq, and Qasr Şirin and its surroundings and Humeyl and Derb Gündeb and its surroundings in Iran. The common feature of these regions is that irrigated agriculture is carried out in the region. The regions in the basin where drought reduction has been detected within the borders of Türkiye are within the scope of the Southeastern Anatolia Project (GAP). GAP is an irrigation project targeting irrigated agriculture to increase the region's economic and social welfare [74]. Similarly, despite all the challenges experienced, the total irrigable land, especially in the north of Syria, is around 1.42 million hectares [75]. Iraq has approximately 13.24 million hectares of land irrigated by the rivers feeding the country within the basin [76]. Iran has approximately 14.3 million hectares of cultivated land under irrigated conditions in the last decade, emphasizing the importance of irrigation in the remaining areas of Iran in the basin [77]. In addition, the region between Baghdad and Basra transitions from the hyperarid class to the arid class. This area is the most important region in central and southern Iraq, where agriculture in the basin mainly depends on irrigation from the Tigris and Euphrates rivers. The region is also within the agricultural areas, expanding with drought threats in Iraq. In Iraq, the agricultural sector consumes the majority of water resources, with values ranging from 75% [78] to 90% [79]. In addition, increasing agricultural activities in the region, based on the amount of water required to protect the existing areas of the marshes in the region, pose

a threat to the sustainable management of the region [80]. In the four countries in the basin, the increase in agricultural areas developed because of approaches to water use and food supply and security is observed in the regions affected by drought. However, it should not be forgotten that these conversion areas are still at risk of drought, and their dependence on water will continue to increase due to the adverse effects of climate change. In addition, when Figure 5d is examined, an increase in drought is observed in the northern and mountainous areas that feed the basin, in Türkiye (Anatolia). This situation is also observed in the drought study conducted in the mountainous areas of Türkiye [81], and another study determined that the Eastern Anatolia Region, which feeds the Euphrates and Tigris basins, is among the four drought hotspots in Türkiye [82]. The increase in droughts observed in the mountainous regions of Türkiye, especially in the northern parts of the Euphrates and Tigris basins, has significant consequences for the hydrology and sustainability of these transboundary river systems. Since these high-altitude areas serve as critical sources of surface water and groundwater through snowmelt and precipitation [83], prolonged drought conditions reduce the amount of water recharged into the main rivers. This may lead to reduced river flows and reduced water availability for agriculture, ecosystems, and downstream communities, and it may prompt more prudent measures among basin countries for the sustainable use of water resources.

Long-term temperature changes across the basin indicate a consistent increase, particularly in the high-altitude regions that geomorphologically serve as the primary sources of water feeding the basin, excluding hyperarid and arid zones (Figure 6a). The analysis of precipitation change reveals a noticeable decline in the northern part of the basin and the southeastern portion of the basin located within the Iranian border (Figure 6c). Both temperature and precipitation change maps show a clear pattern of rising temperatures and decreasing precipitation in the headwater regions, particularly within the Anatolian segment of the basin. The examination of anomaly values supports these findings: yearly temperature anomalies in the mountainous source areas increase up to 1.80 °C (Figure 6b), while yearly precipitation anomalies show a decrease of up to 113.75 mm in the same regions (Figure 6d). Although precipitation anomalies reach deviations of up to 183 mm in the mountain range extending from Anatolia into Iraq and Iran, as well as in the hyperarid and arid zones, their contribution to long-term precipitation trends remains limited. Consequently, these anomalies—characterized by alternating wet and dry years—do not result in a significant long-term increase in precipitation but instead contribute to growing instability in the basin's hydrological cycle. This increased variability underscores the need for enhanced predictability, improved storage capacity, and robust water management systems to ensure resilience in the face of climatic fluctuations. The NEX-GDDP-CMIP6 dataset is comprised of global downscaled climate scenarios derived from the General Circulation Model (GCM) runs conducted under the Coupled Model Intercomparison Project Phase 6 (CMIP6) [53] and across two of the four "Tier 1" greenhouse gas emissions scenarios known as Shared Socioeconomic Pathways (SSPs) [54,84]. This dataset includes downscaled projections from ScenarioMIP model runs [85], for which daily scenarios were produced and distributed through the Earth System Grid Federation. Based on climate projections, the basin is expected to experience significant warming, with temperature deviations reaching +3.05 °C under SSP245 and +6.71 °C under SSP585 by 2100, indicating a strong warming trend regardless of the scenario (Figure 6e). Under SSP245, precipitation is projected to decline slightly by the end of the century, with an average decrease of 8 mm, suggesting increasing aridity. Although SSP585 shows a short-term increase in precipitation by 33 mm around 2040, it declines to only 11 mm above baseline by 2100, indicating a potential for early intensification followed by drying (Figure 6f). These changes imply

heightened drought and water stress risks, along with increased climate variability, which could severely impact the basin's water resources and ecosystem resilience.

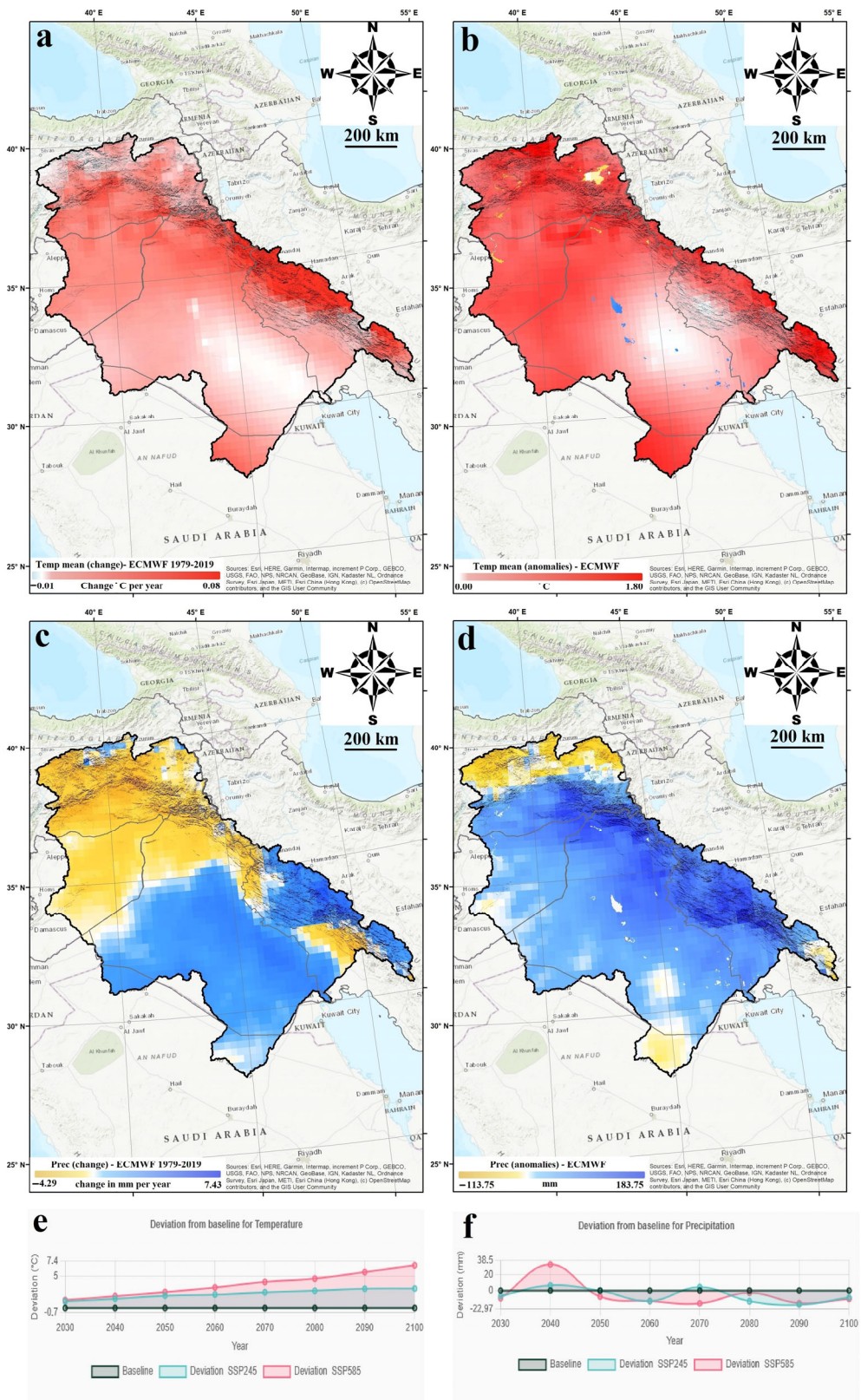

**Figure 6.** Spatial distribution of mean and anomaly temperature and precipitation changes (**a–d**) and temporal trends under SSP245 and SSP585 climate scenarios (**e,f**).

### 3.2. Land Cover and Land Use Change (LCLUC)

An analysis of land use/land cover (LULC) changes from 2000 to 2020 using the GLAD dataset reveals significant spatial and temporal shifts influenced not only by environmental dynamics but also by socioeconomic and political factors across the Euphrates–Tigris Basin. This study focuses on changes in water bodies, cropland, and built-up areas to evaluate the region's sensitivity and adaptive responses to climate change. By analyzing these three critical land cover categories, this study aims to reveal how LULC has evolved in relation to hydrological processes, food security, and urban expansion over the last two decades (Figure 7). An analysis of the 2000–2020 land use/land cover (LULC) changes using the GLAD dataset revealed a decline of 1% in the desert class and 3% in the semi-arid class. Concurrently, a reduction was also observed in certain wetland classes, including salt pan and wetland sparse vegetation classes. In contrast, there was an increase in wetland dense short vegetation and open surface water classes. Notably, the most significant increases were observed in cropland and built-up areas, with expansions of 2.12% and 2.18%, respectively (Table 2). These LULC changes correspond closely with the region's geopolitical instability and development pressures. Countries such as Iraq and Syria have faced prolonged periods of conflict, population displacement, and institutional breakdowns in land governance [86,87]. In many cases, conflict has triggered unregulated agricultural expansion and informal urban development, particularly in peri-urban and semi-arid zones [88]. Cropland and built-up area expansion often reflect a combination of post-conflict reconstruction, internally displaced populations (IDPs) settling in marginal lands, and policies aimed at boosting food security in the face of economic sanctions or conflict-related trade disruptions [89–91].

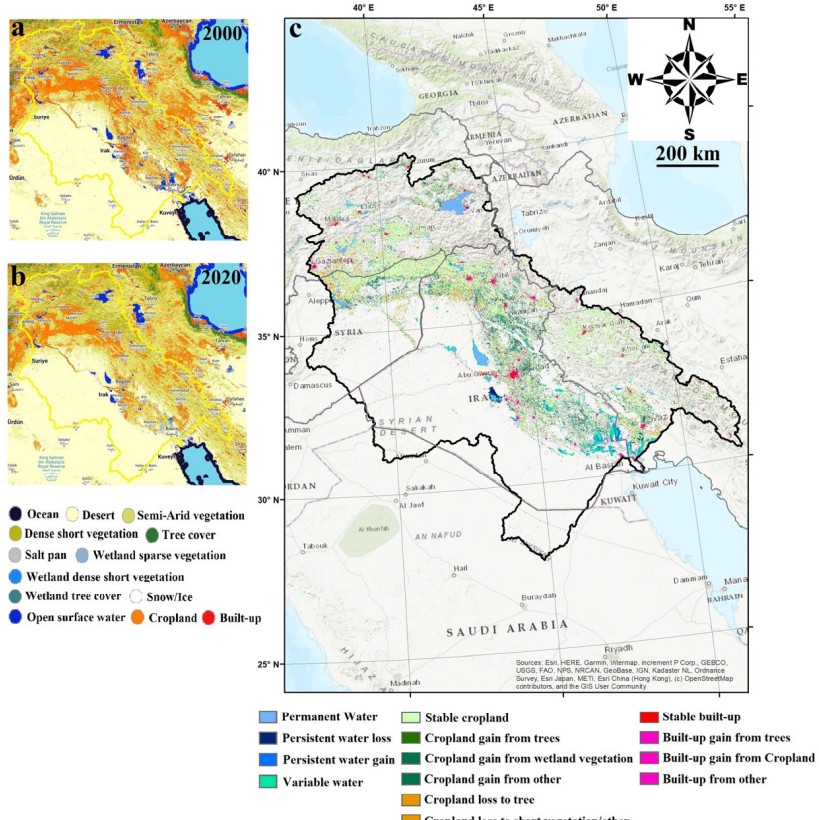

**Figure 7.** The Euphrates–Tigris Basin Land Cover and Land Use Change 2000/2020 UMD GLAD; LULC 2000 (**a**), LULC 2020 (**b**), and LCLUC 2000/2020 (**c**).

**Table 2.** The Euphrates–Tigris Basin Land Cover and Land Use Change.

| Class | 2000 (ha) | 2020 (ha) | Change (ha) |
|---|---|---|---|
| True desert | 37,335,028.45 | 36,484,760.37 | −850,268.08 |
| Semi-arid | 32,850,054.75 | 29,814,243.69 | −3,035,811.06 |
| Dense short vegetation | 6,107,403.91 | 5,793,714.89 | −313,689.02 |
| Tree cover | 225,247.12 | 247,365.85 | 22,118.73 |
| Salt pan | 435,916.26 | 126,023.29 | −309,892.97 |
| Wetland sparse vegetation | 545,022.67 | 167,500.93 | −377,521.74 |
| Wetland dense short vegetation | 99,406.31 | 142,627.95 | 43,221.64 |
| Wetland tree cover | 2726.20 | 2,712.03 | −14.16 |
| Open surface water | 1,321,320.21 | 2,120,124.75 | 798,804.54 |
| Snow/ice | 367.23 | 305.14 | −62.09 |
| Cropland | 13,611,024.57 | 15,598,467.94 | 1,987,443.37 |
| Built-up | 1,003,854.82 | 3,039,525.67 | 2,035,670.85 |
| Ocean | − | − | − |
| Total | 93,537,372.50 | 93,537,372.50 | 0.00 |

According to the GLAD (2000–2020) Land Cover and Land Use Change (LCLUC) matrix assessment, available on Earth Map, wetland loss resulted from land cover conversions, including 1.3% from desert and semi-arid, 6% from cropland, and 1.3% from built-up classes. Conversely, wetland gains were attributed to conversions from desert and semi-arid (6.4%) and cropland classes (5.6%). However, gains in wetland area in desertified regions may be influenced by seasonal flooding or the spread of unregulated irrigation, often driven by short-term adaptation strategies rather than long-term sustainability. Other conversions related to gain and loss areas occurred within the same classes. Cropland loss resulted from land cover conversions, including 7.5% from desert, 75.8% from semi-arid, 11.9% from dense vegetation, and 4.8% from wetland classes. Conversely, cropland gains were attributed to conversions from desert (10%), semi-arid (80.3%), dense vegetation (7.9%), and wetland (1.8%). The expansion of agricultural lands into semi-arid zones, while partly driven by climate adaptation needs, also reflects socioeconomic stress and shifts in land tenure due to displacement and migration. As expected, there are no losses in the built-up class. Built-up gains were attributed to conversions from desert (4.9%), semi-arid (58%), dense vegetation (11.7%), wetland (0.8%), and cropland classes (24.6%). LCLUC main classes and change matrix changes as loss/gain show that cropland and built-up classes are expanding at the expense of semi-arid, desert, dense vegetation, and wetland classes. The results of the drought analysis emphasize that agricultural and urban development pressures are intensifying in the face of drought threats in the basin. Although wetlands have experienced losses due to encroachment from cropland and built-up classes, they have shown gains from desert, semi-arid, and cropland conversions. However, wetland gains, especially in desert and semi-arid classes, are sensitive in terms of water management sustainability due to their conditions. Similarly, cropland dynamics are characterized by significant losses and gains in both semi-arid and desert areas, indicating complex interactions between land degradation and agricultural expansion. The irreversible nature of built-up expansion, driven mainly by conversions from cropland and semi-arid classes, further emphasizes the increasing footprint of urbanization in the basin. In addition, these patterns suggest significant urban growth—much of it unplanned or informal—especially around cities affected by war, population returns, or refugee resettlement. The expansion of cropland into fragile semi-arid and desert zones—such as those seen in Iraq and Syria—can also be interpreted as a result of increased reliance on marginal lands due to war-related infrastructural damage or lack of access to more fertile areas.

In addition, the GLAD 2000–2020 land cover change map, derived from Earth Map, provides classifications of loss, gain, and stability for wetland, cropland, and built-up land cover types. In this study, these changes were analyzed within a GIS environment at the national level, excluding Jordan and Saudi Arabia, in terms of area size and impact on consequences (Table 3). The dataset defines "persistent water loss/gain" for wetland areas; "loss/gain from trees, wetland vegetation, short vegetation, and other" for cropland; and "loss/gain from trees, cropland, and other" for built-up areas. The analysis reveals that 94% of the persistent water loss in the region occurred in Iraq, while persistent water gains were concentrated primarily in Iraq (48.4%) and Syria (40.6%). In terms of cropland expansion, Syria and Iraq recorded the highest rates of cropland gain from wetland vegetation, at 43.9% and 33.4%, respectively. Iraq also exhibited the highest cropland gain from the "other" category (57.6%), which predominantly comprises semi-arid and desert areas. Regarding cropland loss, Iraq and Syria experienced the highest proportions of cropland loss to tree cover, at 44.8% and 42.2%, respectively, while Iraq also had the highest cropland loss to short vegetation or other classes (45.5%). Indicating instability in land use likely tied to abandonment, soil degradation, or land tenure uncertainty. Built-up area expansion was most prominent in Iraq and Türkiye, particularly regarding conversion from cropland. The highest built-up gain from cropland was recorded in Iraq (38.9%), followed by Türkiye (34.7%) and Syria (15.3%). These dynamics highlight the irreversible nature of urban sprawl, which often intensifies in post-conflict periods, such as in Syria and Iraq, and the lack of formal planning or implementation mechanisms. The national-level analysis of land cover dynamics between 2000 and 2020 reveals Iraq as the most affected and transformative country in the region, exhibiting dominant trends across all major land cover transitions

**Table 3.** Land cover percentage gain/loss conversions in the Euphrates–Tigris Basin by country 2000–2020 (except Jordan and Saudi Arabia).

|  | IRN (%) | IRQ (%) | SYR (%) | TUR (%) |
|---|---|---|---|---|
| Persistent water loss | 0.025 | 0.949 | 0.011 | 0.013 |
| Persistent water gain | 0.406 | 0.483 | 0.0103 | 0.099 |
| Cropland gain from trees | 0.247 | 0.29 | 0.009 | 0.452 |
| Cropland gain from wetland vegetation | 0.438 | 0.334 | 0.019 | 0.207 |
| Cropland gain from other classes | 0.140 | 0.575 | 0.107 | 0.176 |
| Cropland loss to trees | 0.422 | 0.447 | 0.011 | 0.119 |
| Cropland loss to short vegetation/other classes | 0.152 | 0.455 | 0.165 | 0.226 |
| Built-up gain from trees | 0.062 | 0.514 | - | 0.423 |
| Built-up gain from crops | 0.152 | 0.389 | 0.111 | 0.346 |
| Built-up gain from other classes | 0.169 | 0.476 | 0.061 | 0.281 |

### 3.3. Vegetation Dynamics, Climatic Water Demand, and Land Degradation Neutrality

Through the Earth Map platform, analyses were conducted on NDVI, PET, and Water Deficit for the Euphrates–Tigris Basin (Figure 8). Since approximately three-quarters of the basin lies within an arid ecosystem, the average NDVI value across the region is relatively low, around 0.18 (Figure 8(a1)). The analysis of NDVI anomaly and change maps indicates that the anomaly values effectively capture vegetation dynamics within the basin (Figure 8(a2,a3)). Notably, significant anomalies and changes are particularly evident in cropland areas. The long-term NDVI trend from 2000 to the present consistently increases, suggesting an overall positive trajectory in vegetation productivity (Figure 8(a4)). The analysis of average PET values reveals that the highest levels are concentrated in the region between Basra and Baghdad, located between the Euphrates and Tigris Rivers, and in areas within the Iranian borders to the north-northeast of Basra (Figure 8(b1)). While the PET anomaly and change maps exhibit broadly similar spatial patterns, the anomaly values effectively reflect the magnitude and distribution of change. Areas showing notable PET variations in both maps include the northern parts of the basin, particularly across

Syria and Türkiye, and the mountainous regions in the eastern and northeastern sectors (Figure 8(b2,b3)). These areas exhibit a pronounced increasing trend in PET values. Despite these localized increases, the overall annual average PET trend across the basin remains relatively stable over the observation period, indicating a generally horizontal trajectory (Figure 8(b4)). According to the average Water Deficit values for the Euphrates–Tigris Basin, notably high deficits are observed in the regions of Baghdad and Basra, as well as in the intensively cultivated and wetland-rich areas located between the two rivers and the zone extending from Kirkuk to Dezful in the eastern part of the basin (Figure 8(c1)). These elevated values indicate that water demand significantly exceeds available supply, particularly in these regions, signifying drought conditions, increased water stress for vegetation, and intensifying pressure on diminishing soil moisture. Additionally, a moderate increase in Water Deficit change is observed in the northern part of the basin, particularly in the Anatolian region of Türkiye, which serves as the basin's headwaters (Figure 8(c2)). This trend, consistent with other findings, points to a worsening drought scenario in the upper basin. Water Deficit anomaly values reinforce this pattern, highlighting a similar spatial distribution (Figure 8(c3)). Notably, a partial decrease in both Water Deficit change and anomaly values is evident in the Zagros Mountains region, stretching from Erbil to Shahr-e Kord in the eastern basin. The long-term trend indicates a marked increase in climatic Water Deficit, particularly pronounced in the period following 2020 (Figure 8(c4)), underscoring the growing impacts of climate change on the regional water balance.

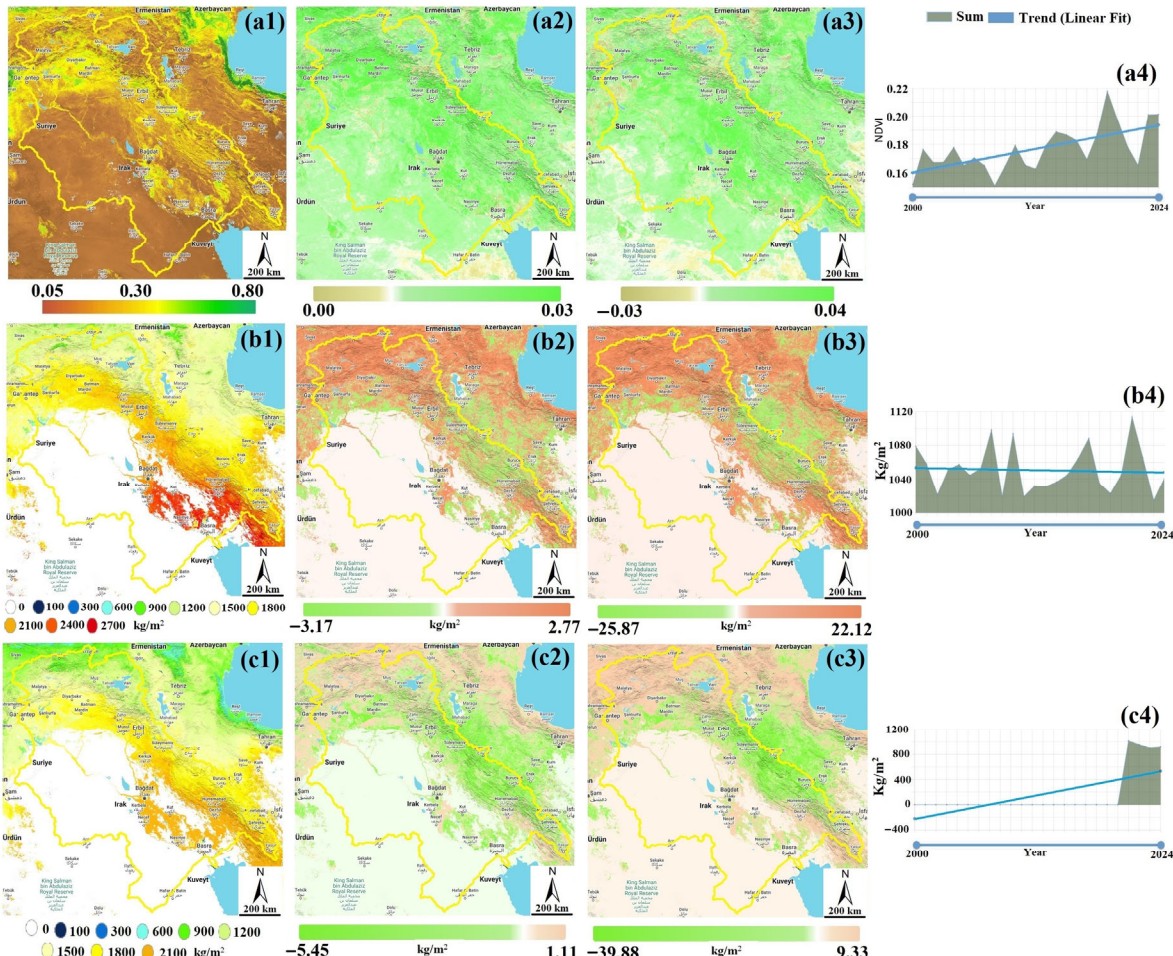

**Figure 8.** Earth Map NDVI, PET, and Water Deficit analysis from 2000 to 2024: NDVI average (**a1**), change (**a2**), anomalies (**a3**), and trend (**a4**); PET average (**b1**), change (**b2**), anomalies (**b3**), and trend (**b4**); and Water Deficit average (**c1**), change (**c2**), anomalies (**c3**), and trend (**c4**).

In this study, LPD was mapped for the Euphrates–Tigris Basin, and the results were analyzed (Figure 9). When analyzing the LPD change matrix for the Euphrates–Tigris Basin from 2016 to 2024 (Table 4), a 66.18% decrease in areas classified as 'Declining' and a 67.39% increase in 'Increasing' areas are observed. This trend is reflected in the class transitions, where 'Increasing' productivity is predominantly detected in cropland areas and their immediate surroundings. An examination of the change matrix and class-specific dynamics related to the 'Cropland' category on Earth Map, within the context of Land Productivity Dynamics, revealed that areas 'Cropland declining' decreased by 44.98%, while those identified as 'Cropland increasing' expanded by 8.89%. These rates of change suggest that the general trend of land productivity decline has slowed in larger areas of cropland, with a concomitant increase in agricultural productivity in areas where land is covered. The observed patterns reflect both a slowdown in the widespread degradation of cropland and the emergence of productivity gains in specific areas.

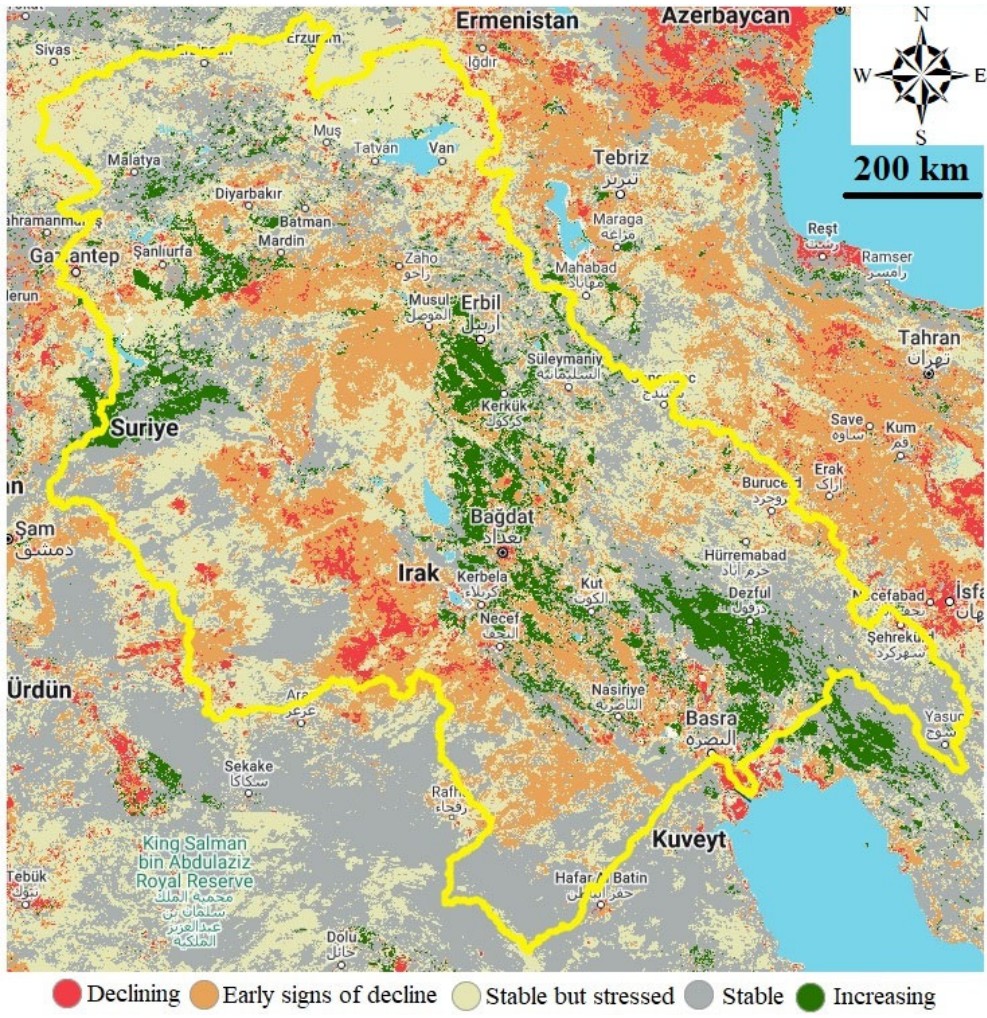

**Figure 9.** The Euphrates–Tigris Basin Land Productivity Dynamics from 2016 to 2024.

In contrast, areas exhibiting a decline in productivity are primarily concentrated in the southwestern parts of the basin, characterized by hyperarid conditions and sparse vegetation cover. Notably, there is a significant increase of 177.01% in the 'Early Signs of Decline' class, which predominantly occurs within the desert and steppe ecozones. This trend suggests a high probability of future degradation, as the largest transition to the 'Declining' category—amounting to 12,561 km$^2$—originates from the 'Early Signs of Decline' class. These regions are known to be globally influenced by sand and dust

storms [92], indicating a potential intensification of dust source activity in the basin over time. Furthermore, the 'Stable but Stressed' class increased by 44.67%, indicating potential vulnerability and its possible contribution to land degradation under worsening conditions. Considering that 55.3% of the basin currently falls within the 'Early Signs of Decline' and 'Stable but Stressed' categories, this supports the hypothesis that these areas, particularly within the desert and steppe ecozones, are at high risk of transitioning into sand and dust source regions. Additionally, the largest transitions to the 'Increasing' class were observed from the 'Stable but Stressed' (53,609.2 km$^2$) and 'Stable' (20,056.22 km$^2$) classes, suggesting localized improvements potentially linked to land management or climatic variability.

**Table 4.** The Euphrates–Tigris Basin Land Productivity Dynamics change matrix (ha).

| | | 2024 | | | | | | Total (2016) |
|---|---|---|---|---|---|---|---|---|
| | | Unknown Class | Declining | Early Signs of Decline | Stable But Stressed | Stable | Increasing | |
| 2016 | Unknown class | 0 | 0 | 0 | 5 | 0 | 0 | 5 |
| | Declining | 0 | 831,411 | 2,063,137 | 3,284,661 | 2,629,552 | 618,859 | 9,427,620 |
| | Early signs of decline | 0 | 1,256,100 | 3,627,536 | 2,654,530 | 2,090,146 | 731,739 | 10,360,051 |
| | Stable but stressed | 19 | 527,488 | 11,849,019 | 10,210,230 | 13,707,808 | 5,360,920 | 41,655,484 |
| | Stable | 0 | 474,596 | 9,503,291 | 5,866,428 | 7,602,693 | 2,005,622 | 25,452,630 |
| | Increasing | 14 | 98,930 | 1,655,257 | 1,031,840 | 1,455,806 | 2,399,523 | 6,641,370 |
| Total (2024) | | 33 | 3,188,525 | 28,698,240 | 23,047,694 | 27,486,005 | 11,116,663 | 93,537,160 |

## 4. Conclusions

This study provides a comprehensive assessment of climate trends, Land Use and Land Cover Change (LULCC), vegetation dynamics, and water resource variability in the Euphrates–Tigris Basin using Earth Map and remote sensing data. The findings highlight critical environmental shifts and their implications for sustainable water management, ecosystem resilience, and regional adaptation strategies under climate change.

The analysis revealed both encouraging and concerning trends in land productivity, with a decline in 'Declining' areas and an increase in 'Increasing' zones, particularly near croplands. However, the sharp rise in 'Early Signs of Decline' and 'Stable but Stressed' areas—especially in hyperarid and steppe ecozones—signals growing degradation risks, potentially leading to expanded dust source regions. These changes are driven not only by climate factors but also by unsustainable land use and land cover practices. Climate trends from ERA5 and MODIS data indicate a shift from hyperarid to semi-arid and dry subhumid zones, largely due to extensive irrigation across Türkiye, Syria, Iraq, and Iran. While such practices have supported localized productivity, they also increase water dependency and vulnerability under projected warming and drying conditions. Rising temperatures and declining precipitation in headwater regions threaten future runoff and basin-wide hydrological stability. Climate projections under the SSP2-4.5 and SSP5-8.5 scenarios further confirm intensifying aridity and extremes. These findings highlight the urgent need for integrated water governance, climate adaptation, and sustainable land management to ensure ecological and socioeconomic resilience in the Euphrates–Tigris Basin.

The analysis of Land Use and Land Cover Change (LULCC) in the Euphrates–Tigris Basin (2000–2020) reveals significant shifts in wetlands, croplands, and built-up areas, largely driven by climatic pressures and human activity. Cropland expansion into semi-arid and desert areas not only reflects growing food demand and adaptation to aridity but also raises concerns about land degradation in environmentally sensitive zones. Built-up area growth, particularly in Iraq and Türkiye, has led to irreversible land conversion and increased stress on water and infrastructure. Wetland changes show both gains and losses,

with many declines linked to cropland and urban encroachment, compounded by past political and security challenges in the southern basin. Iraq stands out as the most affected country, with major transitions in water, cropland, and urban land cover. These findings highlight the close links between land use, climate stress, and socioeconomic development, emphasizing the urgent need for coordinated climate-resilient land management policies that align environmental sustainability with human development in the region.

This study's integrated assessment of vegetation dynamics, climatic water demand, and land productivity in the Euphrates–Tigris Basin provides critical insights into the region's ecological health and vulnerability to climate change and land degradation. Long-term NDVI trends show increased vegetation productivity, particularly in cropland areas, suggesting some resilience through intensified agricultural practices. However, this increase may also reflect unsustainable use of marginal lands. While potential evapotranspiration (PET) remains generally stable, localized rises in headwaters indicate growing evaporative demand under warming. Water Deficits are most severe in densely populated agricultural zones, highlighting increasing risks of drought and vegetation stress, especially after 2020. Land Productivity Dynamics (LPD) reveal mixed trends: a 67% rise in increasing productivity areas contrasts with a sharp 177% surge in early decline signs, mainly in desert and steppe regions vulnerable to dust storms. Additionally, a 45% growth in 'Stable but Stressed' areas points to a widespread risk of degradation without targeted intervention. Overall, more than half of the basin faces potential land degradation, emphasizing the need for coordinated land use and water management to achieve Land Degradation Neutrality. Sustainable practices, enhanced restoration efforts, improved water efficiency, and regional cooperation are essential to safeguard the basin's ecological and socioeconomic resilience amid climate change.

The Euphrates–Tigris Basin is undergoing rapid environmental changes driven by natural pressures, climate variability, and anthropogenic pressures. While some regions show adaptive capacity (e.g., cropland expansion), others face escalating water stress and land degradation. Addressing these challenges requires coordinated science-based policies to ensure the basin's long-term sustainable water management, Land Degradation Neutrality, ecosystem stability, and socioeconomic resilience. Earth Map and remote sensing tools have been proven to be invaluable for monitoring these dynamics and informing decision-making in a data-scarce region. While the current results provide valuable insights into the environmental dynamics of the Euphrates–Tigris Basin, they are clearly insufficient to fully grasp the complex socioeconomic impacts faced by downstream communities. In particular, socioeconomic data must also be integrated to fully understand socioeconomic impacts. Integrating these human dimensions is crucial for developing comprehensive and targeted adaptation strategies that enhance basin-wide resilience and promote equitable resource management. Future studies should integrate higher-resolution datasets and localized socioeconomic assessments to refine adaptation strategies. For this purpose, it is important to identify the subheadings of each problem that will increase resilience and contribute to the solution and to produce policies by revealing their effects on each problem.

**Author Contributions:** Conceptualization, A.A., M.H.A. and K.Y.; Methodology, A.A., A.S.-P.D. and G.M.; Investigation, A.A. and F.Ş.B.; Data Curation, A.A. and F.Ş.B.; Writing—Original Draft Preparation, A.A., M.H.A. and K.Y.; Writing—Review and Editing, A.S.-P.D., G.M. and F.Ş.B.; Supervision, A.S.-P.D. and G.M.; Project Administration, A.A. All authors have read and agreed to the published version of the manuscript.

**Funding:** The APC was funded by the Republic of Türkiye Ministry of Environment, Urbanization and Climate Change and the Directorate General for Combating Desertification and Erosion within the framework of the national project entitled "Preparation of Türkiye's Desertification Sensitivity and Risk Maps" which is Project Number: 4389.

**Institutional Review Board Statement:** Not applicable.

**Informed Consent Statement:** Not applicable.

**Data Availability Statement:** The datasets can be found at https://help.earthmap.org/ (accessed on 1 June 2025).

**Acknowledgments:** The authors gratefully acknowledge the Food and Agriculture Organization (FAO) of the United Nations for developing and maintaining the Earth Map platform, which enabled the integrated analysis presented in this study. We also thank the Google Earth Engine team for providing the computational infrastructure that supports large-scale geospatial analysis. Their open-access tools are invaluable for environmental monitoring in data-scarce regions. This research was also conducted within the framework of the national project entitled "Preparation of Türkiye's Desertification Sensitivity and Risk Maps", which provided essential context and data inputs for the assessment.

**Conflicts of Interest:** The authors declare no conflicts of interest.

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
