# Peer review of "Insights from Earth Map: Unraveling Environmental Dynamics in the Euphrates–Tigris Basin"

_sustainability, doi:10.3390/su17167513_

Round 1
Reviewer 1 Report
Comments and Suggestions for Authors
The Euphrates-Tigris basin is a very large region passing through different geological regions it’s very difficult to sum up its environmental considerations in a scientific way in a single paper. Why is such a huge region considered instead of focusing on a particular zone?
Climate projections scenarios SSP245 and SSP585 shall be illustrated in the introduction section how these scenarios interact with the area under study.
Since the Euphrates-Tigris basin comprises three different countries with different socioeconomic conditions, the water sharing agreement(s), their history, and future aspects are vital to discuss.
Some flow patterns reflecting drought and flood conditions for both rivers shall be plotted /tabulated for at least some stations.
The river pollution and its classification are very important if environmental dynamics are required to be studied what type of river water contamination is reported and what are their consequences.
In case of a decrease in overall river discharge, the lower riparian areas are very vulnerable. The socioeconomic effects are devastating for farmers and fishermen, and these issues cannot be easily recognized by using satellite images. What are the recommendations to study the socioeconomic aspects of lower riparians?
The conclusion section is very large; it shall be summed up in a couple of paragraphs. The other discussions and explanations should be moved to the relevant sections, and if necessary, a discussion section can be included. Some discussions were repeated in the conclusion part, which shall be removed.
Author Response
Comment 1 : The Euphrates-Tigris basin is a very large region passing through different geological regions it’s very difficult to sum up its environmental considerations in a scientific way in a single paper. Why is such a huge region considered instead of focusing on a particular zone?
Response 1 : While it is true that the Euphrates-Tigris basin encompasses diverse geological and ecological zones, studying the basin as a whole is essential for understanding the integrated and transboundary nature of environmental and climate-related challenges. The basin functions as a hydrological and ecological unit, where upstream and downstream processes are closely interconnected. Focusing on isolated subregions would risk overlooking the cumulative impacts of land use change, water management, and climate variability that transcend political and ecological boundaries. For instance, alterations in precipitation patterns or land cover in the upstream areas of Turkey can significantly affect water availability, and vegetation dynamics in downstream areas of Syria and Iraq.
Moreover, the comprehensive monitoring of such a large basin aligns with international climate adaptation and mitigation goals, especially under frameworks like the UNFCCC. Modern geospatial platforms such as EarthMap and Google Earth Engine enable the analysis of long-term, large-scale environmental trends using consistent datasets across national borders. This basin-wide approach supports a holistic understanding of the region’s vulnerability to climate change and human pressures, allowing for coordinated policy recommendations, resource management strategies, and adaptation measures. While detailed case studies within the basin are valuable, an overarching view is critical for identifying broad patterns, systemic risks, and opportunities for cooperative action.
Comment 2 : Climate projections scenarios SSP245 and SSP585 shall be illustrated in the introduction section how these scenarios interact with the area under study.
Response 2 : To address the criticism regarding the scale and scope of the study area, climate projection scenarios SSP2-4.5 and SSP5-8.5 have been integrated into the introduction section (Line 84-94). These scenarios are used to illustrate the projected climate trajectories for the Euphrates–Tigris Basin, particularly in terms of temperature and precipitation changes. Relevant references have been included to support the use of these scenarios and to align the study with current climate modeling practices.
Comment 3 : Since the Euphrates-Tigris basin comprises three different countries with different socioeconomic conditions, the water sharing agreement(s), their history, and future aspects are vital to discuss.
Response 3 : We appreciate this insightful critique, as it underscores a critical dimension of transboundary water governance in the Euphrates–Tigris Basin. The historical, political, and socioeconomic complexities surrounding water-sharing among the riparian countries—Türkiye, Syria, and Iraq—are indeed central to understanding the region’s environmental and resource management challenges. These factors were a primary consideration in the selection of the basin as the focus of this study. In response, the scope of the Study Area section has been expanded to address issues related to water use, governance frameworks, and the prospects for future cooperation, with appropriate references to relevant literature and agreements (line 144-155).
By situating these transboundary dynamics within the broader geographic and climatic context of the basin, the study adopts a more integrated perspective that links hydrological patterns, climate variability, and geopolitical realities. This comprehensive approach enhances the justification for focusing on the Euphrates–Tigris Basin and underscores the pressing need for coordinated, climate-resilient strategies for water and land management. It also highlights the importance of fostering regional collaboration to address shared environmental challenges under both current conditions and future climate scenarios.
Comment 4 : Some flow patterns reflecting drought and flood conditions for both rivers shall be plotted /tabulated for at least some stations.
Response 4 : We appreciate the reviewer’s suggestion regarding the inclusion of flow patterns that reflect drought and flood conditions at selected stations along the Euphrates and Tigris Rivers. While we acknowledge the importance of such hydrological analyses in understanding river dynamics, this specific aspect falls outside the scope of the present study, which focuses primarily on basin-wide climate trends, land use changes, and their transboundary implications. Detailed hydrological modeling or station-level flow data analysis represents a distinct line of inquiry that, while complementary, would require a different methodological framework and dataset than those employed in this research.
Nevertheless, we agree that such an investigation could provide valuable insights in future work, particularly in studies dedicated to assessing hydrological extremes and their local impacts. We have noted this point for consideration in the design of subsequent research phases.
Comment 5 : The river pollution and its classification are very important if environmental dynamics are required to be studied what type of river water contamination is reported and what are their consequences.
Response 5 : We thank the reviewer for this valuable comment, which highlights the importance of river pollution and its classification in understanding environmental dynamics. We fully acknowledge that water quality and pollution are critical components of river basin health and sustainability. However, it is important to clarify that the current study primarily focuses on climate-related factors, land use changes, and transboundary resource management in the Euphrates-Tigris Basin. Therefore, the assessment and classification of river pollution goes beyond the specific objectives and methodological scope of this study.
However, we agree that water quality requires comprehensive investigation, particularly given its relevance to ecological integrity, human health, and regional cooperation. We see this as an important avenue for future research and believe that the findings of this study will contribute to subsequent studies on water pollution assessments.
Comment 6 : In case of a decrease in overall river discharge, the lower riparian areas are very vulnerable. The socioeconomic effects are devastating for farmers and fishermen, and these issues cannot be easily recognized by using satellite images. What are the recommendations to study the socioeconomic aspects of lower riparians?
Response 6 : We appreciate the reviewer's insightful commentary on the vulnerability of lower coastal areas to reduced river flows and the resulting socioeconomic impacts on local communities, particularly farmers and fishermen. In light of this important observation, the conclusion has been revised to reflect the need for further research on the socioeconomic dimensions. At the reviewer's suggestion, a recommendation to include socioeconomic data in future studies has been added (line 736-742). This holistic approach is essential to capture the full scope of impacts and develop more comprehensive and inclusive adaptation strategies.
Comment 7 : The conclusion section is very large; it shall be summed up in a couple of paragraphs. The other discussions and explanations should be moved to the relevant sections, and if necessary, a discussion section can be included. Some discussions were repeated in the conclusion part, which shall be removed.
Response 7 : Thank you for your insightful feedback. We acknowledge that the original conclusions section contained excessive detail and some repetitive content. In response, we have refined the results and conclusions to present a more concise summary of the key findings and their implications. The conclusions have been streamlined to clearly emphasize the critical points, enhancing both clarity and coherence (line 683-728). This restructuring ensures a logical flow throughout the manuscript and strengthens the overall presentation. We appreciate your constructive critique, which has greatly contributed to improving the quality of this work.
Reviewer 2 Report
Comments and Suggestions for Authors
1.A brief summary
This paper aims to comprehensively assess the environmental dynamics in the Euphrates-Tigris Basin by analysising climate trends, land use land cover changes (LULCC), vegetation dynamics, and water resource variability in the Euphrates-Tigris Basin using Earth Map. This study provides valuable insights for the region’s ecological health and its vulnerability to climate change and land degradation and highlights the urgent need for integrated water management and climate-resilient policies to sustain the basin's ecological and socio-economic resilience. One of the highlights of this study is the use of Earth Map, which provides democratizes access to environmental data for policymakers and stakeholders, to quickly assess environmental issues across large areas, especially in regions where data is scarce.
- General concept comments
(1) The study effectively leverages Earth Map to provide a holistic analysis of multiple environmental parameters in the Euphrates-Tigris Basin. Although this study integrates numerous climatic factors and spatial environmental variables, including precipitation, temperature, vegetation cover, land use change, NDVI, PET, and water deficit, most of the work merely presents the dynamics of these variables individually. It lacks a quantitative analysis of the relationship between these environmental variables and climate change. In other words, the study does not clearly demonstrate how climate change impacts environmental dynamics based on the data. Additionally, the research objective is not well-defined, and the absence of a technical roadmap further limits the study's clarity and direction.
(2) It is suggested that a table be added in the methods section to list all the data used in this study, along with their sources and purposes.
(3)It is recommended that the data and processing methods described in the Results and Discussions section be moved to the Methods section of this study, where the introduction of the Earth Map platform should be summarized.
3.Specific comments
(1) All the maps(Figure 1-2.,4-9) in the article lack scales and compass directions. Need to be supplemented.
(2)In lines 225–229,the potential evapotranspiration product from the MOD16A2 MODIS/Terra Net Evapotranspiration 8-Day L4 Global 500m version 6 image collection is mentioned for calculating the Aridity index. According to the subsequent content provided by the author,the Aridity index is calculated on an annual basis. However, the text does not provide information on how to aggregate the 8-day potential evapotranspiration product into an annual product. Additionally,there is no information on the spatial resolution of the precipitation data. Further clarification is needed on how these two datasets are matched when calculating the Aridity index.
(3)In lines 503–505,the term“Land Degradation Neutrality(LDN)”appears twice.It is suggested that the abbreviation“LDN”be used alone the second time it appears.
(4)In lines 445–448, the author mentions that the PET data used for calculating the Water Deficit spans the years 2000–2017. However, in Figure 8c4, the trend of Water Deficit is shown from 2000 to 2024, and the sum values only appear close to the year 2024. There is an inconsistency between these two pieces of information. The author is requested to further verify this discrepancy.
Author Response
1.A brief summary
This paper aims to comprehensively assess the environmental dynamics in the Euphrates-Tigris Basin by analysising climate trends, land use land cover changes (LULCC), vegetation dynamics, and water resource variability in the Euphrates-Tigris Basin using Earth Map. This study provides valuable insights for the region’s ecological health and its vulnerability to climate change and land degradation and highlights the urgent need for integrated water management and climate-resilient policies to sustain the basin's ecological and socio-economic resilience. One of the highlights of this study is the use of Earth Map, which provides democratizes access to environmental data for policymakers and stakeholders, to quickly assess environmental issues across large areas, especially in regions where data is scarce.
- General concept comments
Comment 1: The study effectively leverages Earth Map to provide a holistic analysis of multiple environmental parameters in the Euphrates-Tigris Basin. Although this study integrates numerous climatic factors and spatial environmental variables, including precipitation, temperature, vegetation cover, land use change, NDVI, PET, and water deficit, most of the work merely presents the dynamics of these variables individually. It lacks a quantitative analysis of the relationship between these environmental variables and climate change. In other words, the study does not clearly demonstrate how climate change impacts environmental dynamics based on the data. Additionally, the research objective is not well-defined, and the absence of a technical roadmap further limits the study's clarity and direction.
Response 1a : We sincerely appreciate the reviewer’s thoughtful engagement with our work and the constructive feedback provided. In response, we would like to clarify several important aspects of the study that may not have been fully recognized. While the comment suggests an absence of quantitative analysis linking environmental variables to climate change, our approach is firmly grounded in spatial and temporal data analysis. Utilizing Earth Map’s globally standardized datasets, we conducted a comprehensive GIS-based assessment of key environmental indicators—including NDVI, precipitation, temperature, potential evapotranspiration (PET), land use change, and water deficit too. These parameters were not merely presented descriptively; rather, they were quantitatively processed, analyzed, and interpreted to reveal trends and dynamics indicative of climate-related change across the Euphrates-Tigris Basin.
The structure of the paper reflects a clear analytical roadmap through the core sections: Climate Trends and Variability, Land Cover and Land Use Change (LCLUC), and Vegetation Dynamics and Climatic Water Demand & Land Degradation Neutrality. Each section is designed to build upon the previous, offering a progressively integrative analysis of the region’s environmental trajectory. These sections incorporate trend analyses and spatial overlays that enable quantitative assessments of change over time, grounded in remote sensing-derived indices and metrics. While we acknowledge that deeper inferential statistical modeling (e.g., regression or machine learning approaches) could further elucidate causal relationships, the primary objective of this study was to establish a robust baseline of environmental dynamics—a foundational step for future hypothesis-driven or model-intensive research.
Importantly, the study's research design is aligned with the intended use of Earth Map as a platform for large-scale, policy-relevant environmental monitoring. Earth Map’s value lies in its ability to aggregate, standardize, and visualize complex geospatial data, enabling interdisciplinary analyses across regions and time periods. Our work demonstrates how such a platform can be leveraged not only for thematic mapping but also for structured analytical investigations. The transparent, reproducible nature of Earth Map’s data architecture reinforces the credibility of our findings and supports their applicability to regional climate adaptation and land management strategies.
In this context, our study prioritizes a multi-parameter, systems-based approach aimed at identifying spatial vulnerabilities and environmental stress patterns under evolving climate conditions. The identification of these patterns is, in our view, a necessary precursor to any attempt at modeling causality or projecting future scenarios. We fully agree that further research—especially studies incorporating more advanced statistical or process-based modeling—would be valuable and are in fact a logical extension of the groundwork laid here.
In summary, we respectfully maintain that the study offers a structured, quantitatively grounded, and thematically coherent analysis that contributes to both scientific understanding and practical decision-making. We trust that this clarification helps to more accurately convey the scope, rigor, and intention of our work.
Response 1b: We thank the reviewer for the helpful observation regarding the clarity of the research objective. In response, we have revised the final paragraph of the introduction to more clearly articulate the study’s purpose (Line 120-130). The updated statement now defines the aim as assessing the impacts of climate change on environmental dynamics in the Euphrates-Tigris Basin through the integration of geospatial indicators using Earth Map. This revision strengthens the focus and relevance of the study, aligning it with its intended contribution to evidence-based resource management and climate adaptation. We believe this revision addresses the reviewer’s concern and provides a more focused and clearly articulated research objective within the broader context of climate change and resource management.
Comment 2: It is suggested that a table be added in the methods section to list all the data used in this study, along with their sources and purposes.
Response 2: We appreciate the reviewer’s constructive suggestion to include a table listing all the data used in the study, along with their sources and purposes. In response, we have added Table to the Methods section, which provides a detailed summary of the datasets, including their sources and intended use within the study (Line 385-386). We believe this addition improves the clarity and transparency of the methodology.
Comment 3 : It is recommended that the data and processing methods described in the Results and Discussions section be moved to the Methods section of this study, where the introduction of the Earth Map platform should be summarized.
Response 3: We appreciate the reviewer’s insightful recommendation regarding the organization of the manuscript. In response, we have moved the relevant descriptions of data and processing methods from the Results and Discussion section to the Methods section, where they are more appropriately placed. Additionally, the introduction to the Earth Map platform has been revised and summarized to maintain focus and clarity in the Methods section (Line 224-384)
3.Specific comments
Comment 1: All the maps(Figure 1-2.,4-9) in the article lack scales and compass directions. Need to be supplemented.
Response 1 : We thank the reviewer for this valuable observation. In response, all maps presented in Figures 1–2 and 4–9 have been revised to include scale bars and north arrows to enhance their readability and geographic orientation (Line 156, 220, 406, 415,.499, 525, 639, 657). These additions aim to improve the interpretability and cartographic quality of the visual materials in the manuscript.
Comment 2: In lines 225–229,the potential evapotranspiration product from the MOD16A2 MODIS/Terra Net Evapotranspiration 8-Day L4 Global 500m version 6 image collection is mentioned for calculating the Aridity index. According to the subsequent content provided by the author,the Aridity index is calculated on an annual basis. However, the text does not provide information on how to aggregate the 8-day potential evapotranspiration product into an annual product. Additionally,there is no information on the spatial resolution of the precipitation data. Further clarification is needed on how these two datasets are matched when calculating the Aridity index.
Response 2: Thank you for your valuable feedback. We acknowledge the need for greater clarity regarding the processing steps involved in calculating the Aridity Index (AI) and the spatial matching of the datasets used. We revised the manuscript to include these methodological details for improved transparency and reproducibility (Line 249-262)
Comment 3: In lines 503–505,the term“Land Degradation Neutrality(LDN)”appears twice.It is suggested that the abbreviation“LDN”be used alone the second time it appears.
Response 3: Thank you for your observation. As suggested, the repeated use of the full term Land Degradation Neutrality (LDN) has been revised. In line 280, only the full term is used, and in the other lines, the abbreviation "LDN" is applied accordingly.
Comment 4: In lines 445–448, the author mentions that the PET data used for calculating the Water Deficit spans the years 2000–2017. However, in Figure 8c4, the trend of Water Deficit is shown from 2000 to 2024, and the sum values only appear close to the year 2024. There is an inconsistency between these two pieces of information. The author is requested to further verify this discrepancy.
Response 4: Thank you for bringing this to our attention. Upon review, it was confirmed that the PET data used for calculating the Water Deficit, as presented in Figure 8c4, indeed spans the period from 2000 to 2024, based on the data available through Earth Map. The reference to the 2000–2017 period in the line was a typographical error and has been corrected accordingly to ensure consistency with the data shown in the figüre (Line 338).
Reviewer 3 Report
Comments and Suggestions for Authors
Comment 1: In my opinion, the main problem is that the authors rely too heavily on the Earth Map platform without critically analyzing its limitations or comparing it to other analytical methods. The study lacks independent validation of the results using ground-based data or alternative sources. To assess the accuracy of the analysis, the authors should include a comparison of "Earth Map" results with data from ground-based weather stations, hydrological stations, and alternative satellite products.
Comment 2: Additionally, the authors provide only a basic description of the research methods used, which is somewhat lacking in depth. They are advised to specify the algorithms used to process time series, the methods used to calculate trends, the statistical tests used to determine the significance of changes, the formula used to calculate the aridity index, the methods used to spatially interpolate climate data, and so on.
Comment 3: The authors present data on LULC changes based on the GLAD dataset, but their interpretation of the causes of these changes is insufficient. They did not consider socio-economic factors, political instability in the region, or the impact of conflicts on land use, and they are recommended to address these potential drivers.
Comment 4: (L243-253) The claim by the authors that irrigation projects are the main driver of reduced aridity in the region is questionable, as it does not take into account natural climate variability and possible errors in ERA5-Land data in arid regions.
Comment 5: The maps presented have insufficient resolution for publication in a scientific journal. Figures 5-9 are especially problematic because the legends are difficult to read. Additionally, the color schemes are not optimal for differentiating between different types of classes.
I wish that my comments would be helpful in improving the quality of this research.
Thank you.
Author Response
Comment 1: In my opinion, the main problem is that the authors rely too heavily on the Earth Map platform without critically analyzing its limitations or comparing it to other analytical methods. The study lacks independent validation of the results using ground-based data or alternative sources. To assess the accuracy of the analysis, the authors should include a comparison of "Earth Map" results with data from ground-based weather stations, hydrological stations, and alternative satellite products.
Response 1 : Thank you for this thoughtful and important comment. We agree that validation using ground-based data is critical in many environmental studies and appreciate the opportunity to clarify the scope and limitations of our methodology. The study area in question—the Euphrates–Tigris Basin—spans several countries and encompasses a wide range of climatic and topographic conditions. Given its geopolitical complexity, including ongoing conflict and instability in parts of the region, access to reliable and consistent ground-based weather or hydrological station data is severely constrained. Where such data do exist, they are often inaccessible to the scientific community due to national security restrictions or data-sharing limitations, and in some cases, their reliability and consistency over long time series are questionable.
The primary aim of this study is to provide a broad, basin-scale perspective on climate-induced changes in hydrological and ecological conditions over the past two decades. For this purpose, Earth Map was selected as the main analytical platform due to its integration of several validated global datasets and its accessibility for large-scale, long-term monitoring. Earth Map enables the use of harmonized satellite-derived products (e.g., MODIS, CHIRPS, ERA5-Land), offering consistent temporal and spatial coverage that would not be feasible to achieve through fragmented ground station networks across the basin. The tool has been developed in collaboration with FAO and the Google Earth Engine platform, and relies on peer-reviewed data sources, many of which are commonly used in global and regional climate and land monitoring studies.
Nonetheless, we acknowledge that Earth Map, like any remote sensing-based platform, has inherent limitations, particularly in terms of local-scale precision and potential algorithmic uncertainties. These limitations are explicitly recognized in the revised manuscript, and readers are cautioned that the outputs should be interpreted in terms of spatial and temporal trends rather than point-specific absolute values. In the absence of dense and consistent ground data, Earth Map offers a viable and transparent alternative for deriving first-order insights into environmental changes across politically sensitive or data-scarce regions.
While we were unable to perform direct validation using in-situ measurements, we have taken care to cross-reference and contextualize our findings with results from previous peer-reviewed studies conducted in parts of the basin where data were available. For example, observed patterns in vegetation decline, changes in precipitation, and water deficit trends are broadly consistent with existing literature, lending additional confidence to our interpretation of the Earth Map outputs.
In future work, we fully agree that the inclusion of ground-based validation—should access and data availability improve—will be critical for refining and confirming the satellite-derived results. Additionally, comparison with alternative satellite products (e.g., TRMM, SMAP, GRACE, or Landsat-based indices) may further enhance the robustness of the findings, and such efforts are already being considered for the next phase of this research.
In conclusion, while the use of Earth Map does present certain limitations, its adoption in this study is based on practical constraints and methodological suitability for large-scale, transboundary analysis. The approach is intended not as a substitute for ground-based data, but as a complementary method for visualizing and understanding spatial patterns of change across vast and complex landscapes. We have updated the manuscript to more clearly acknowledge these limitations and to transparently communicate the rationale behind our methodological choices (Line 224-386)
Comment 2: Additionally, the authors provide only a basic description of the research methods used, which is somewhat lacking in depth. They are advised to specify the algorithms used to process time series, the methods used to calculate trends, the statistical tests used to determine the significance of changes, the formula used to calculate the aridity index, the methods used to spatially interpolate climate data, and so on.
Response 2 :Thank you for your valuable comment. We appreciate your suggestion to expand the methodological detail and fully agree that providing more comprehensive information on data processing and analytical techniques strengthens the transparency and reproducibility of the study. The Methods section has been revised and enriched to include the specific algorithms, statistical approaches, and formulas used throughout the analysis. These methodological clarifications have been integrated into the revised manuscript to address the comment comprehensively. We thank the reviewer for encouraging a more rigorous presentation of the analytical framework (Line 224-386).
Comment 3: The authors present data on LULC changes based on the GLAD dataset, but their interpretation of the causes of these changes is insufficient. They did not consider socio-economic factors, political instability in the region, or the impact of conflicts on land use, and they are recommended to address these potential drivers.
Response 3 : We thank the reviewer for this insightful and important comment regarding the interpretation of land use/land cover (LULC) changes. In response to the suggestion, we have carefully revised the manuscript to expand our discussion of the underlying drivers of LULC dynamics, particularly by incorporating socio-economic factors, political instability, and the impacts of regional conflicts.
In the revised version of the manuscript, additional paragraphs have been added to the relevant section (please see line 503-524, 534-536, 541-543, 558-563)) to contextualize the observed LULC changes—such as cropland and built-up expansion, wetland and vegetation loss—within broader socio-political processes in the region. These additions aim to provide a more comprehensive interpretation of the observed trends and align the analysis more closely with the complex human-environment interactions affecting the region.
We appreciate the reviewer’s valuable recommendation, which has significantly improved the depth and clarity of our interpretation.
Comment 4: (L243-253) The claim by the authors that irrigation projects are the main driver of reduced aridity in the region is questionable, as it does not take into account natural climate variability and possible errors in ERA5-Land data in arid regions.
Response 4 :We appreciate the reviewer's concern regarding the potential influence of natural climate variability in drylands and the accuracy and precision of ERA5-Land data in arid regions. It is indeed well-established that natural climate variability plays a significant role in shaping drought patterns, especially in dryland environments where precipitation is highly variable and extreme conditions are common. However, the regions where we observed a decrease in drought coincide with areas characterized by large-scale irrigation projects. These areas show a consistent correlation between the presence of irrigation infrastructure and reduced drought severity, suggesting that irrigation has been a contributing factor to the observed drought mitigation.
With respect to the use of ERA5-Land data, we acknowledge that there may be limitations in highly localized and arid settings (Yilmaz, M. (2023). Accuracy assessment of temperature trends from ERA5 and ERA5-Land. Science of The Total Environment, 856, 159182.). Nonetheless, ERA5-Land remains one of the most widely used and validated sources of climate and meteorological data, offering a spatial resolution (9 km) that is generally appropriate for regional-scale studies (Muñoz-Sabater, J., Dutra, E., Agustí-Panareda, A., Albergel, C., Arduini, G., Balsamo, G., ... & Thépaut, J. N. (2021). ERA5-Land: A state-of-the-art global reanalysis dataset for land applications. Earth system science data, 13(9), 4349-4383.). This makes it particularly suitable for basin-wide assessments, such as those conducted in the Euphrates and Tigris basins. While we recognize the justified critiques of ERA5 performance in smaller-scale or highly heterogeneous arid regions, we encourage future research to incorporate ground-truthing or higher-resolution datasets where available to improve the accuracy of local-scale analyses (Xu, J., Ma, Z., Yan, S., & Peng, J. (2022). Do ERA5 and ERA5-land precipitation estimates outperform satellite-based precipitation products? A comprehensive comparison between state-of-the-art model-based and satellite-based precipitation products over mainland China. Journal of Hydrology, 605, 127353.).
Overall, we believe the evidence supporting the role of irrigation in drought mitigation is robust—especially when analyzed alongside vegetation dynamics (e.g., NDVI trends), irrigation expansion data, and hydrological indicators. While we fully consider the role of natural climate variability in drylands, the spatial consistency of drought reduction in regions with known irrigation infrastructure supports our conclusion that irrigation has played an important role in moderating aridity in these specific areas.
Comment 5: The maps presented have insufficient resolution for publication in a scientific journal. Figures 5-9 are especially problematic because the legends are difficult to read. Additionally, the color schemes are not optimal for differentiating between different types of classes.
Response 5 :We thank the reviewer for their feedback regarding the figure resolution and color schemes. All figures, including Figures 5-9, were submitted at a resolution appropriate to the journal's submission guidelines. We understand that during the review process, the page layout and scaling may affect the clarity and legibility of the figures, particularly the legends and detailed class representations. However, we are confident that, upon final formatting for publication, the resolution and readability will meet the required standards.
We also acknowledge the reviewer’s concern regarding the complexity of the color schemes. The use of a wide range of colors was necessary due to the high number of land use/land cover (LULC) classes and the spatial complexity of class transitions. We carefully selected color palettes that maximize contrast and minimize confusion between similar classes while adhering to cartographic standards. In addition, all figures are accompanied by detailed legends and with supporting tables and descriptions in the manuscript to ensure clarity for the reader.
We appreciate our reviewer's attention to visual clarity and are confident that the final figures will meet the required standards of resolution and readability after final formatting. We also send the all orginal figures to journal.
Round 2
Reviewer 2 Report
Comments and Suggestions for Authors
The article has been revised and has basically answered the questions I raised, but there are still some issues that need attention:
1. The method section introduces too much about the environmental variable data used, but lacks an introduction to the analysis method for the dynamic changes in the environmental variable data. For example, the results and discussion sections mention the anomalies of environmental variables multiple times (e.g., Figure 8), but the method section does not explain how these anomalies are defined.
2. The titles of some figures are not very clear. For instance, the titles of Figure 8 and Figure 9 should include the specific time.
3. The semicolons in lines 226, 249, 263, 286, 313, and 349 are not very appropriate.
Author Response
Comment 1 : The method section introduces too much about the environmental variable data used, but lacks an introduction to the analysis method for the dynamic changes in the environmental variable data. For example, the results and discussion sections mention the anomalies of environmental variables multiple times (e.g., Figure 8), but the method section does not explain how these anomalies are defined.
Response 1 : Thank you for this insightful and constructive feedback. We fully acknowledge the validity of the concern regarding the methodological description of the dynamic analysis applied to the environmental variable data. In the revised manuscript, we have carefully integrated the environmental variable datasets—each introduced as distinct subheadings in the Methods section—with the corresponding Results and Discussion content. Specifically, we have elaborated on the rationale for selecting these variables and clarified how they collectively contribute to addressing the core objectives of the study (Lines 250-253; 268-274; 298-306; 334-342; 381-392; 426-436) These additions ensure a clearer methodological framework that transparently connects data processing steps to the subsequent findings. We believe these enhancements significantly improve the clarity and rigor of the methodology, thereby strengthening the overall coherence and reproducibility of the study
Comment 2. The titles of some figures are not very clear. For instance, the titles of Figure 8 and Figure 9 should include the specific time.
Response 2 : Thank you for your valuable critique regarding the clarity of the figure titles. We have addressed this by explicitly adding the specific time periods to the titles of both Figure 8 and Figure 9 (Line 695 and 713). This revision ensures that the temporal context of the data presented is clearly conveyed to the reader, enhancing the interpretability and precision of the visual information.
Comment 3. The semicolons in lines 226, 249, 263, 286, 313, and 349 are not very appropriate.
Response 3: Thank you for your careful attention to detail regarding the use of semicolons in lines 226,227, 254, 275, 307, 343, and 393. We have reviewed and corrected these instances in accordance with the journal’s formatting guidelines. Additionally, the sections have been renumbered following the third-level heading structure as required. We appreciate your helpful suggestions in improving the manuscript’s clarity and consistency.
Note : The manuscript has been carefully reviewed and revised to improve clarity, coherence, and overall English language quality
Reviewer 3 Report
Comments and Suggestions for Authors
The manuscript has improved significantly since the last revision, and in my opinion, it can be published in its current form
Author Response
Comment 1 : The manuscript has improved significantly since the last revision, and in my opinion, it can be published in its current form
Response 1: Dear Reviewer,
We sincerely thank you for your thorough review and thoughtful comments throughout the revision process. Your valuable insights and constructive feedback have greatly contributed to improving the quality and clarity of our manuscript.
We truly appreciate the time and effort you dedicated to evaluating our work, and we are grateful for your positive assessment and recommendation for publication.
Thank you once again for your support and encouragement.
Kind regards,